# $^1$H $R_{1\rho}$ relaxation identifies a hidden intermediate in DNA base-pairing

Rubin Dasgupta [1,2,4], Christian Steinmetzger[1,2,4], Julian Ilgen[1,3] & Katja Petzold [1,2] ✉

$^1$H $R_{1\rho}$ Relaxation dispersion (RD) NMR experiments provide valuable atomic-level insights into transient, high-energy conformational states of biomolecules. However, cross-relaxation artifacts can hamper its interpretation and therefore limiting broader adoption. This study explicitly quantifies cross-relaxation effects on $^1$H $R_{1\rho}$ relaxation rates, extending the general applicability of $^1$H $R_{1\rho}$ to probe dynamics at natural abundance. Artifacts were found to be negligible for neighbouring dipolar-coupled protons, >3 Å apart, and a concept for identification for protons less than 3 Å is provided. This approach revealed a second excited state (ES2) in DNA base-pairing that extends the well-established Watson-Crick-Franklin (WCF) ground state (GS) − Hoogsteen (HG) equilibrium. A structural model for ES2 is proposed based on evidence from $^1$H $R_{1\rho}$ RD, trapping via DNA modifications, metadynamics simulations, and DFT-based chemical shift calculations. ES2 was stabilised by the anticancer drug Actinomycin D, providing direct experimental evidence that small molecule can remodel conformational landscape of DNA. Together, these results demonstrate both a methodological advance by establishing reliable conditions for $^1$H $R_{1\rho}$ RD studies, and a mechanistic discovery of a drug-stabilised intermediate in DNA base-pairing dynamics.

Conformational dynamics in biomolecules play a crucial role in defining their biological function[1–3]. In these biomolecules, microsecond to millisecond time scale dynamics are present between an energetically favourable ground state (GS) and a higher energy, excited state (ES)[4–6]. ES conformations have been reported to have important roles in nucleic acids, e.g. microRNA processing[7,8], targeting[1], DNA base repair[9], and HIV activation[10]. Due to their low population (typically <2%)[4,5], ESs are challenging to characterise using either X-ray crystallography or cryo-electron microscopy. NMR spectroscopy, however, has proven to provide atomic-resolution structural and dynamical information about these ESs[6,10–12].

Measuring the longitudinal relaxation rate in the rotating frame ($R_{1\rho}$) and its dispersion with respect to the applied spinlock field strength ($\omega_{SL}$; on-resonance) and spinlock offset relative to the resonance of interest ($\Omega_{SL}$; off-resonance) has been the method of choice to identify such ES[6]. $R_{1\rho}$ relaxation dispersion (RD) quantifies the conformational exchange contribution ($R_{ex}$) to the transverse relaxation rate ($R_2$) of the resonance under study[6,13]. It can probe exchange rates between 50 Hz and 50 kHz and provide access to the chemical shift of the ES[6,14–16]. This chemical shift information allows modelling the structure of an ES[12].

The ES conformation typically observed in DNA is the Hoogsteen (HG) base-pair, which is crucial in processes like DNA damage repair[17,18], and recognition of DNA by transcription factors[19,20]. In an HG base-pair, the purine nucleobase adopts a *syn* conformation around the glycosidic bond (C1′−N9) rather than *anti* as in a Watson-Crick-Franklin (WCF) (Fig. 1a, b)[21,22]. $^{13}$C and $^{15}$N RD experiments have been employed to characterise the HG ES in DNA[2,3,6,9,11,23–28]. Recently, a $^1$H

[1]Department of Medical Biochemistry and Biophysics, Karolinska Institutet, Stockholm, Sweden. [2]Department of Medical Biochemistry and Microbiology, Center of Excellence for the Chemical Mechanisms of Life & Science for Life Laboratory, Uppsala University, Uppsala, Sweden. [3]Present address: Institute of Organic Chemistry, University of Regensburg, Regensburg, Germany. [4]These authors contributed equally: Rubin Dasgupta, Christian Steinmetzger. ✉e-mail: katja.petzold@imbim.uu.se

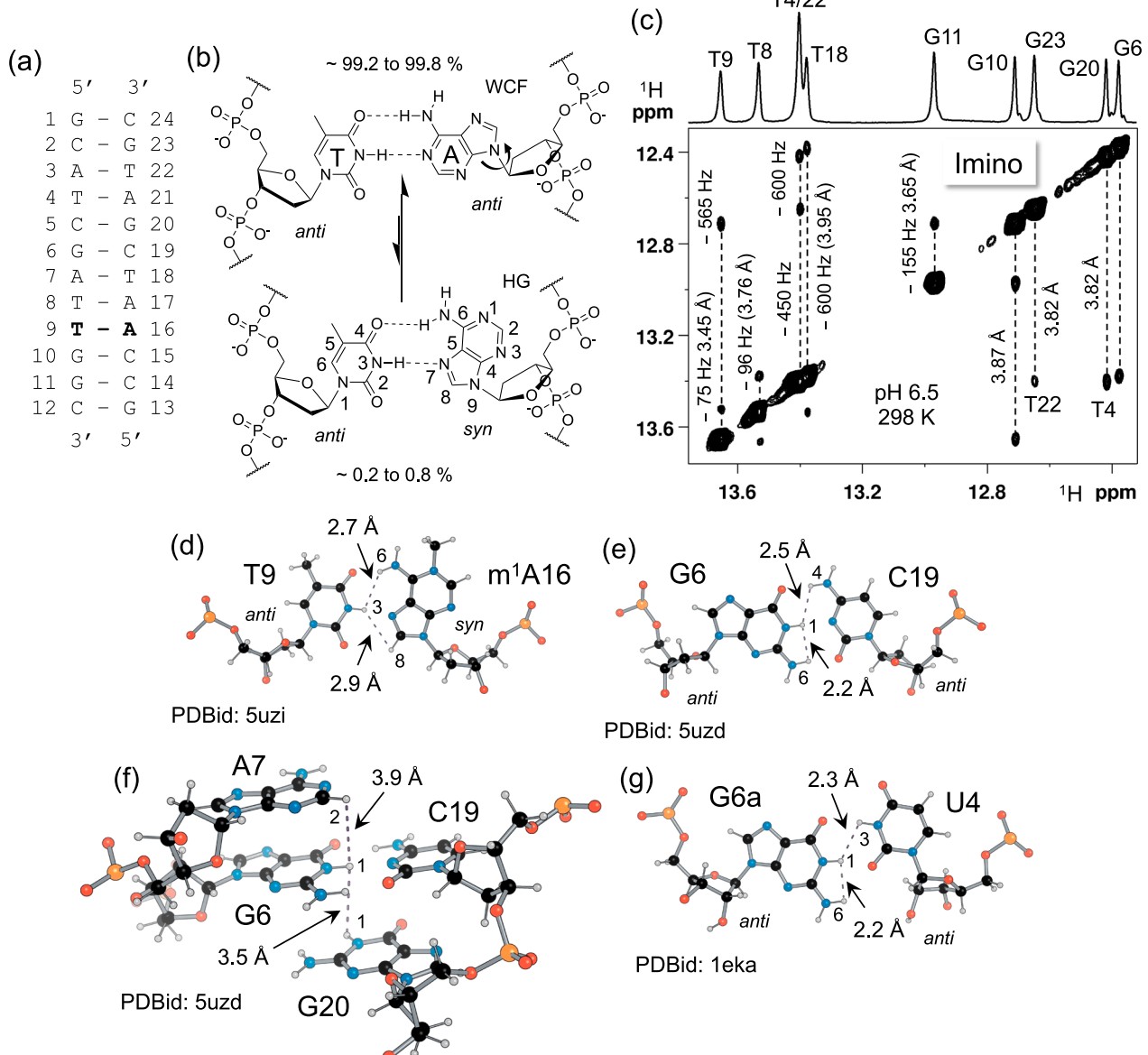

**Fig. 1 | Model system used in this study. a** A$_2$ DNA sequence[3] with the base-pair T9–A16 marked in bold, where a representative WCF–HG dynamics was measured. **b** WCF–HG conformations for the T9–A16 base-pair with the reported populations and *anti*-to-*syn* transition for A16[3,21,24]. **c** Selective imino NOESY[68] spectrum of A$_2$ DNA. The 1D projection is shown at the top of the spectrum with the resonance assignment[21,24]. The position of the cross-peaks (in Hz, relative to the diagonal signal) and the corresponding average distance from the 10 minimum energy structures (in Å) in PDBid 5uzd[30] are denoted. Representative NMR-derived intra- and inter-base-pair proton–proton distances that may contribute to ¹H $R_{1\rho}$ relaxation rates: **d** T9–m¹A16 HG base-pair in A$_6$ DNA, PDBid 5uzi[30], **e** G6–C19 WCF base-pair in A$_2$ DNA, PDBid 5uzd[30], **f** Base-pair steps from G6–C19 to the neighbouring A7 and G20 in A$_2$ DNA, PDBid 5uzd[30] and **g** U4–G6 wobble base-pair in r(GAGUGCUC)$_2$ RNA, PDBid 1eka [https://doi.org/10.2210/pdb1EKA/pdb][80]. Atomic position numbers are provided for protons involved in hydrogen bonds.

high-power chemical exchange saturation transfer (CEST) experiment was developed to study WCF–HG dynamics[21] in a model DNA oligonucleotide, A$_2$ DNA (Fig. 1a)[3,21]. The exchange rates that can be probed with these methods, however, are limited to $k_{ex}$ up to 4–5 kHz[6]. The ¹H $R_{1\rho}$ RD experiment expands the accessible exchange rate to tens of kHz, is more sensitive compared to CEST[14,15] and can be readily applied to systems at natural isotopic abundance. Additionally, ¹H chemical shifts are particularly sensitive towards identifying non-canonical base-pairs conformation in nucleic acids[29].

¹H $R_{1\rho}$ RD for a given proton might, however, have contributions from neighbouring protons due to cross-relaxation that could be mistaken for additional conformational exchange[14]. An NMR structure ensemble of A$_2$ DNA[30] indicates ᴺH–ᴺH distances in the range of 3.4–3.9 Å between consecutive base-pairs (Fig. 1c). Distances from ᴺH

to other protons within a base-pair in typical nucleic acid geometries range from 2.2 Å to 2.9 Å in various structural contexts in both DNA and RNA (Fig. 1d–g). These distances might pose a challenge to obtain reliable exchange parameters from ¹H $R_{1\rho}$ RD experiments on A$_2$ DNA.

This study shows that cross-relaxation between ᴺH protons at typical inter-base-pair distances as described above has negligible impact on $R_{1\rho}$ RD profiles, which consequently can be analysed using well-established models for conformational exchange[6,31–34]. A significant effect is seen at shorter interproton distances (Fig. 1d, e, g), which can be mitigated by prudent choice of spinlock strengths and offsets in the $R_{1\rho}$ experiment. A thorough theoretical description of cross-relaxation effects on ¹H $R_{1\rho}$ RD using Bloch-McConnell matrix propagation[35] is discussed under different exchange scenarios. In A$_2$ DNA, this allowed the detection of a previously underappreciated

excited state, ES2, involved in the WCF–HG equilibrium. To explore the structural implications of ES2, the HG-forming adenine in the central T9–A16 base-pair was chemically modified to obtain chemical shift fingerprints of different base-pair conformations[12]. This ES2 is also shown to be promoted in presence of an anti-tumour drug Actinomycin D. The well-tempered parallel bias metadynamics simulation[36–38] in combination with chemical shift calculation a structural model for the ES2 is proposed. Overall, the discovery of ES2 offers a deeper understanding of the energy landscape of the WCF–HG transition, which may ultimately provide significant insights into its role in DNA biochemistry.

## Results and discussions

### Effects of cross-relaxation on $^1H$ $R_{1\rho}$ relaxation dispersion

The commonly assumed issue of cross-relaxation on $^1H$ $R_{1\rho}$ RD profiles is assessed for a two-state GS, ES exchange model including dipolar coupling to a neighbouring proton at variable distances (Fig. 2a, b, e, g). Both auto-relaxation and cross-relaxation contributions were included to evaluate their effects on on-resonance and off-resonance $^1H$ $R_{1\rho}$ RD experiments. Simulations indicate a distance dependence (Fig. 2c, d, f, h). For nearest-neighbour proton distances ≥3 Å, characteristic of inter-base-pair imino–imino contacts in canonical DNA and RNA (Fig. 1f), cross-relaxation contributes ≤5% to the observed $^1H$ $R_{1\rho}$ rates (Fig. 2c, f). Under these conditions, $^1H$ $R_{1\rho}$ RD profiles are well described by standard chemical-exchange models, enabling reliable extraction of exchange parameters.

At shorter interproton distances (<2.5–3 Å, blue in Fig. 2c, d, f profiles), representative of geminal sugar protons and closely spaced intra-base-pair contacts (Fig. 1d, e, g), cross-relaxation leads to elevated on-resonance $R_{1\rho}$ rates (Fig. 2c, f, h). An additional maximum at the position of the dipolar-coupled proton is observed in off-resonance profiles (Fig. 2d). These effects are only observed when the chemical shift of the neighbouring proton falls within the sampled spinlock offset range. Simulations in which cross-relaxation is present in both GS and ES, or cross-relaxation is larger in the ES, yield similar trends but with a slightly more restrictive distance threshold for negligible effects (5% below ~3.2 Å; Fig. 2e–h). In all scenarios, cross-relaxation effects are minimal when neighbouring protons are ≥3.2 Å away and the chemical shift of the resonances lie outside the probed offset window. A thorough description of the effect of cross-relaxation at different exchange regimes and for anisotropic molecule is explained in the supporting information (Supplementary Figs. 1 and 2)

Together, these results define practical conditions under which $^1H$ $R_{1\rho}$ RD experiments can be analysed using established exchange models without artifacts. Exchange measurements on nucleobase protons are therefore robust for typical nucleic acid geometries, provided that neighbouring resonances are considered during experimental design, easily identifiable via Nuclear Overhauser Effect.

### $^1H$ $R_{1\rho}$ RD reveals third state in WCF–HG transition

$^{13}C$ and $^{15}N$ $R_{1\rho}$ as well as high-power $^1H$ CEST experiments on $A_2$ DNA have demonstrated that WCF–HG dynamics follow a two-state exchange model with an exchange rate of 3–4 kHz and HG population of 0.2–0.8%[3,21,24]. Motivated by the above results of manageable cross-relaxation effects, $^1H$ $R_{1\rho}$ RD experiments were conducted to study the WCF–HG transition in $A_2$ DNA. The NOESY spectrum of the imino $^NH$ region shows that the dipolar-coupled T9 $^NH3$–T8 $^NH3$ and T9 $^NH3$–G10 $^NH1$ pairs are at chemical shifts of −75 Hz (−0.12 ppm) and −565 Hz (−0.94 ppm), respectively, relative to T9 $^NH3$ (Fig. 1c). Given their ~ 3.9 Å distance[30], the dipolar interactions between the proton pairs minimally influence the on- and off-resonance $^1H$ $R_{1\rho}$ rates (Fig. 2c, d).

$^1H$ $R_{1\rho}$ RD data for T9 $^NH3$ revealed two conformational exchange contributions (Fig. 3a, Supplementary Fig. 3a and Supplementary Table 1) modelled using a triangular three-state exchange topology

(Supplementary equation 20) and statistically preferred over all other tested models by F-test, Akaike Information Criterion, and Bayesian Information Criterion[6,39] (Supplementary Data 1). Comparing with the previous reported chemical shifts[21], the conformation at $\Delta\omega = -593 \pm 10$ Hz ($-0.99 \pm 0.02$ ppm) corresponds to the HG state ($\Delta\omega_{HG}$), with exchange rate ($k_{ex, WCF \leftrightarrows HG}$) = 2.7 ± 0.15 kHz and population ($p_{HG}$) = 0.6 ± 0.01% (Fig. 3a, b, Supplementary Table 1). Additionally, a second excited state (ES2) at $\Delta\omega_{ES2} = +288 \pm 9$ Hz ($+0.48 \pm 0.01$ ppm) with $p_{ES2}$ = 0.9 ± 0.1% was observed. The exchange rates with ES2 are $k_{ex, WCF \leftrightarrows ES2}$ = 0.5 ± 0.07 kHz and $k_{ex, HG \leftrightarrows ES2}$ = 3.2 ± 0.15 kHz (Fig. 3b and Supplementary Table 1). This ES2 represents an intermediate during the WCF–HG transition in $A_2$ DNA. Furthermore, the system displays a temperature-dependent change in topology. A linear topology dominated at 283, 288, and 308 K, while a triangular topology was observed at 293, 298, and 303 K (Supplementary Fig. 4). This behaviour, coupled with the non-linear van't Hoff plot (Supplementary Fig. 4b) and as previously observed[3], likely arises from the presence of ES2, thereby making the derivation of reliable thermodynamic parameters challenging.

$^1H$ $R_{1\rho}$ RD experiments on the base-pairing partner of T9, A16, which undergoes the *anti*-to-*syn* transition to form the HG conformer, further confirmed the presence of ES2. Non-exchangeable aromatic protons $^CH2$ and $^CH8$ were globally fitted to a three-state linear topology (Supplementary Fig. 3, Supplementary Table 1 and Supplementary Data 1), sharing $k_{ex, WCF \leftrightarrows HG}$, and $p_{HG}$ values. However, their sensitivity to WCF–ES2 transitions was limited, resulting in higher uncertainty in the estimation of the ES2 population. Consequently, a global fit combining T9 and A16 $^1H$ $R_{1\rho}$ RD data was not feasible due to differing exchange rates and populations (Supplementary Table 1). Other protons in the neighbouring T–A base-pairs (T8 $^NH3$, T18 $^NH3$, and A17 $^CH2$) also indicated the presence of an ES2 (Supplementary Fig. 3). T8 $^NH3$ and A17 $^CH2$, ES2 were possible to be globally fitted to a linear three-state exchange model with a shared $k_{ex, HG \leftrightarrows ES2}$, and $p_{ES2}$, without resolved WCF–ES2 contributions (Fig. 3b and Supplementary Table 1). T18 $^NH3$ showed three-state star-like exchange where only the WCF–ES2 transition was resolved, with no HG–ES2 transition detected. The reason for the associated comparatively high $p_{ES2}$ (12.2 ± 0.9%) and low $k_{ex, WCF \leftrightarrows ES2}$, (24 ± 0.7 Hz) could not be explained with the current dataset. This suggests that certain triangular exchange components may remain unresolved in some situations. Nevertheless, the observed $p_{HG}$ and $\Delta\omega_{HG}$ exchange parameters for T9, T8, and T18 $^NH3$ match the previous report using high-power $^1H$ CEST[21] (Fig. 3b), while the difference in $k_{ex, WCF \leftrightarrows HG}$ can be attributed to different pH values of the sample used in the present study. Overall, these findings establish that ES2 is a consistent intermediate state within T–A base-pairs in $A_2$ DNA, reflecting the complex dynamics of WCF–HG transitions.

### Structural information of ES2 by chemical modification of A16

To gain insight into the conformation of ES2, three $A_2$ DNA constructs with chemically modified nucleobases at A16 were prepared (Fig. 4a, Supplementary Tables 2 and 3): (i) Purine ($A_2$ P16) substitution to eliminate the amino group that hydrogen-bonds with O4 of T9[40], reducing base-pair strength in both WCF and HG conformations; (ii) 7-deazaadenine substitution ($A_2$ c$^7$A16), which removes the N7 hydrogen bond acceptor, favouring the WCF conformation by sterically interfering with HG base-pair formation[24]; and (iii) 1-methyladenine substitution ($A_2$ m$^1$A16), where methylation of N1 promotes formation of the HG conformation by blocking the WCF edge of A[3]. Chemical shift perturbations (CSPs) of imino protons due to these modifications are localised primarily to the T9–A16 base-pair and its adjacent ±1 neighbours (Fig. 4b). NOESY and SOFAST HMQC spectra confirmed T9 WCF base-pairing with $A_2$ P16 and $A_2$ c$^7$A16, and HG base-pairing with $A_2$ m$^1$A16 (Supplementary Figs. 5–7), consistent with prior reports[24,25,30,40,41]. Imino proton ($^NH3/1$) chemical shifts showed that in the GS conformation, T9 $^NH3$ of both $A_2$ P16 and $A_2$ m$^1$A16 is shifted

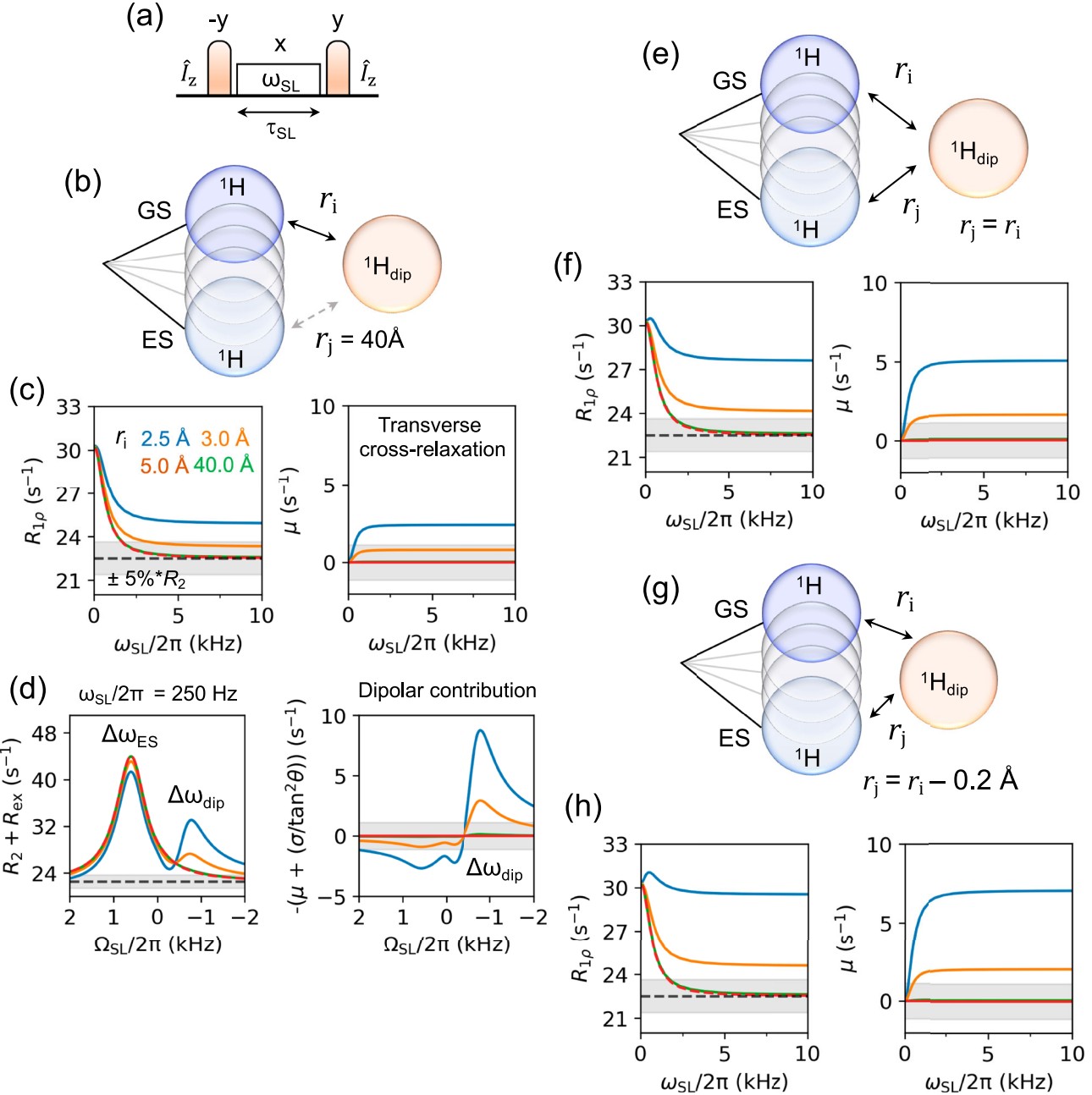

**Fig. 2 | Effects of cross-relaxation on $^1H\,R_{1\rho}$ RD experiments. a** $R_{1\rho}$ pulse sequence used in simulations: Z magnetisation on spin $I$ ($I_Z$) is rotated to the ZX plane by a $-y$ pulse, spinlock with strength $\omega_{SL}$ for $\tau_{SL}$, and returned to Z-axis with a $y$ pulse. **b** Two-site chemical-exchange model between ground state (GS) and excited state (ES) (blue circles), with a dipolar-coupled proton, $^1H_{dip}$ (light brown). Distances between $^1H_{dip}$ and GS is denoted as $r_i$ and $r_j$, respectively where $r_j = 40$ Å representing scenario 1. **c** On-resonance profile (left) for $r_i = 2.5, 3.0, 5.0$ and 40 Å (blue, orange, brown, and green) depicting that at for $r_i \geq 3$ Å, cross-relaxation effects remain within ±5% error (grey region) of the $R_2$ rate, ensuring accurate extraction of exchange parameters. Transverse cross-relaxation ($\mu$) rate contribution (right) to the on-resonance $R_{1\rho}$ profile shows the same trend. **d** Off-resonance $R_2 + R_{ex}$ profile (left) for scenario 1 at $\omega_{SL}/2\pi = 250$ Hz with $\Delta\omega_{ES}$ and $\Delta\omega_{dip}$ depicted in the plot. This

shows that the dipolar proton has a response at its $\Delta\omega_{dip}$ at ≤3 Å. The dipolar contribution on the off-resonance profile (right) shows that there is a substantial contribution on the response from ES ($\Delta\omega_{ES}$) at distances <3 Å, complicating the extraction of reliable exchange parameters. **e, g** Two-state exchange model where $r_j = r_i$ or $r_j = r_i - 0.2$ Å representing scenarios 2 and 3, respectively. **f, h** On-resonance profile (left) and contribution from $\mu$ (right) shows that cross-relaxation in the ES amplifies $R_2$ contribution ($\mu$) more in scenarios 2 and 3 than in scenario 1 where the distance to the $^1H_{dip}$ must be >3 Å to be within ±5% error range of $R_2$. Parameters used for the simulations are $k_{ex} = 2$ kHz, $p_{ES} = 0.5\%$, $\tau_c = 5.1$ ns, $R_{1GS} = R_{1ES} = R_{1dip} = 2.5$ s$^{-1}$, $R_{2GS} = R_{2ES} = R_{2dip} = 22.5$ s$^{-1}$, $\Delta\omega_{ES}/2\pi = +600$ Hz and $\Delta\omega_{dip}/2\pi = -600$ Hz. The grey band in (**c, d, f, h**) represents ±5% assumed error of the average calculated $R_2$ rate (22.50 ± 1.13 s$^{-1}$), representing typical experimental error.

upfield, while in A$_2$ c$^7$A16 it is shifted downfield relative to the wild-type DNA (A$_2$ wt) (Fig. 4c). This deshielding in A$_2$ c$^7$A16 was attributed to the altered electrostatic properties of the c$^7$A modification, increased solvent accessibility, and changes in stacking interactions[24,41]. Coincidentally, the T9 $^NH3$ shift in A$_2$ c$^7$A16 matched that of ES2 in A$_2$-wt (Fig. 4c), a state with a lower base-pairing stability.

Using $^1H\,R_{1\rho}$ RD at 278 K in A$_2$ P16, the T9 $^NH3$, P16 $^CH2$ and P16 $^CH8$ signals fitted a three-state linear exchange model (Fig. 4d, Supplementary Fig. 8 and Supplementary Data 1). The fitted $\Delta\omega_{HG}$ conformation has the same sign as the HG-trapped A$_2$ m$^1$A16 DNA, while the $\Delta\omega_{ES2}$ followed the sign observed in A$_2$ wt (Fig. 3b), supporting this assignment. A$_2$ P16 showed increased $k_{ex, HG \leftrightarrows ES2}$ (18 ± 0.6 kHz) and

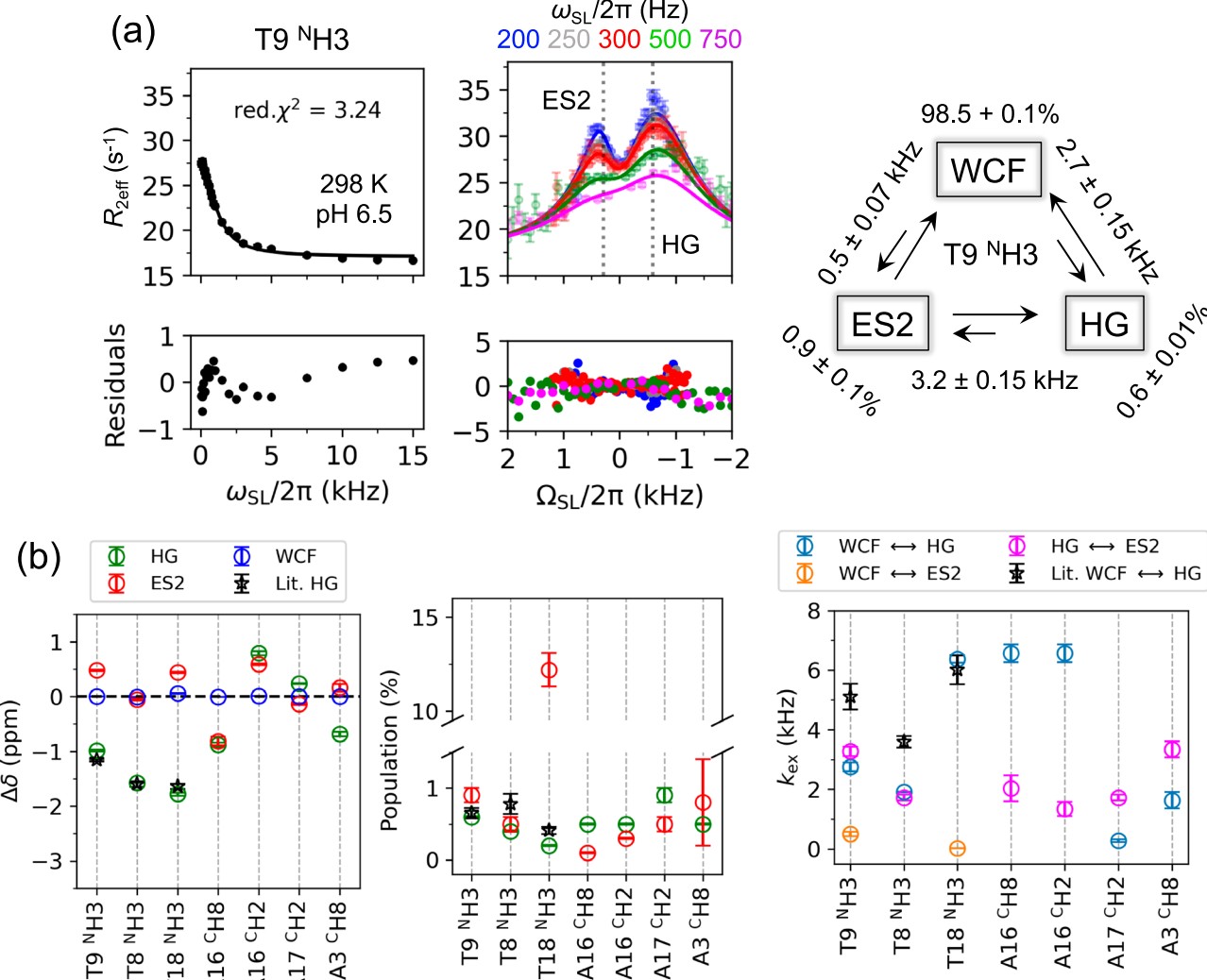

**Fig. 3 | ¹H $R_{1\rho}$ RD reveals ES2 in WCF–HG dynamics in $A_2$ DNA. a** On-resonance (left) and off-resonance (right) $R_{2eff}$ plots for T9 $^NH3$ at 298 K and pH 6.5 with a three-state exchange fit. Solid lines represent the fits associated with the fit parameters (Supplementary Table 1), and dashed lines show relative chemical shifts of HG and ES2. Exchange rates and populations for each conformer are shown in the schematic representation of the best fitted three-state triangular exchange model. Reduced $\chi^2$ (red. $\chi^2$) is denoted on the on-resonance plot and the associated residuals for both on- and off-resonance are shown below the plots. $R_{2eff}$ error bars represent ±1 SD calculated from 500 Monte Carlo iterations of fits to the underlying monoexponential signal decays and noise levels of the corresponding NMR spectra. **b** Relative Chemical shifts ($\Delta\delta$ (ppm)) of HG (green) and ES2 (red) with respect to WCF (blue) are shown for each studied atom. $\Delta\omega_{HG}$ and $p_{HG}$ from ¹H $R_{1\rho}$ RD of $^NH3$ in T9, T8, and T18 are consistent with the literature data from high-power ¹H CEST[21] (black star). This confirms ES2 as an intermediate state during the WCF–HG transition. The error bars were estimated from Monte Carlo sampling of 500 replicas, representing mean ±1 SD. Source data are provided as a Source data file.

$p_{ES2}$ (19.3 ± 2.2%) (Fig. 4d, Supplementary Table 4 and Supplementary Table 5) compared to $A_2$ wt (Supplementary Fig. 4a and Supplementary data 1). This suggests that the increased flexibility of the T9–P16 base-pair enhances ES2 sampling. Similar exchange patterns observed for P16 $^CH8$ and P16 $^CH2$ (Fig. 4d, Supplementary Fig. 8, Supplementary data 1) indicate that the findings are not confounded by water exchange on the imino proton, despite lower relative stability of the T9–P16 base-pair[42]. The 20-fold increase in $p_{ES2}$ and 10-fold increase in $k_{ex, HG \leftrightarrows ES2}$ for $A_2$ P16 with respect to $A_2$ wt underscore the role of the amino group in limiting ES2 sampling.

In $A_2$-c$^7$A16, ¹H $R_{1\rho}$ RD at 278 K for T9 $^NH3$ also revealed a three-state linear exchange (Fig. 4d, and Supplementary Fig. 8) with one excited state at $\Delta\omega = -1748 \pm 27$ Hz ($-2.91 \pm 0.04$ ppm), consistent with the HG conformation observed in other DNA duplexes[3,21,43]. This is intriguing since the c$^7$A modification sterically disfavours the HG conformation, suggesting this is an HG-like state that requires further investigation. The other excited state exhibited $\Delta\omega = 127 \pm 89$ Hz

(+0.21 ± 0.13 ppm), resembling the ES2 in $A_2$ wt. The exchange parameters for this ES2-like state in $A_2$-c$^7$A16 were comparable to those in $A_2$-wt (Supplementary Fig. 4a, and Supplementary Data 1), indicating that N7 does not influence the ES2 conformation as significantly as the N6 amino group, as shown in $A_2$-P16. Notably, this exchange in c$^7$A-modified DNA was undetected in earlier ¹³C and ¹⁵N $R_{1\rho}$ studies[24], highlighting the enhanced sensitivity of ¹H $R_{1\rho}$ RD.

For $A_2$-m$^1$A16 DNA, ¹H $R_{1\rho}$ RD on T9 $^NH3$ at 278 K exhibited a dispersive on-resonance profile, indicative of a minor conformation with $k_{ex} > 25$ kHz. The rapid exchange precluded precise estimation of population and $\Delta\omega$ values from off-resonance experiments (Supplementary Fig. 7e and Supplementary Table 4).

## Actinomycin D binding promotes ES2 conformation

To investigate the relevance of ES2 in presence of a known DNA-binding drug, $A_2$ wt DNA was treated with the cytostatic compound Actinomycin D (ActD). Although the $A_2$ DNA does not contain the

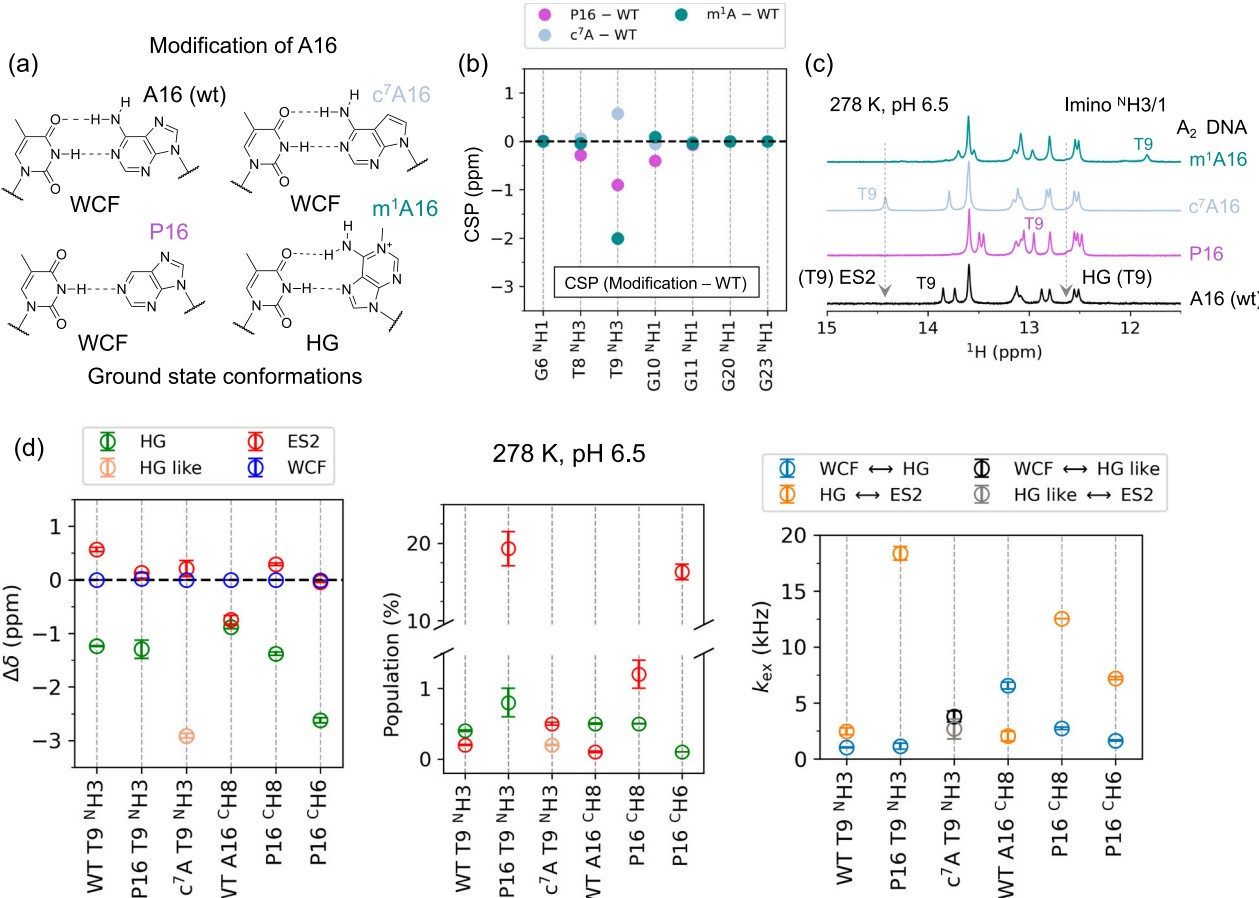

**Fig. 4 | Effects of A16 modification on WCF–HG–ES2 dynamics. a** Chemical structures of unmodified (A2-wt) and modified A16 nucleobases (A2-P16, -c⁷A16, and -m¹A16) used to modulate the WCF–HG exchange equilibrium in A2 DNA. **b** Chemical shift perturbations (CSPs) of imino protons in modified A2 DNA constructs show localised effects near the site of modification (T9–A16 base-pair), with minimal influence on adjacent base-pairs and no detectable perturbation beyond ±1 neighbours. **c** ¹H imino spectra at 278 K and pH 6.5 for A2-wt (black), A2-P16 (magenta), A2-c⁷A16 (steel blue), and A2-m¹A16 (dark green), highlighting the position of the T9 ᴺH3 resonance. Arrows indicate the chemical shift corresponding to the minor ES2 and HG conformations as identified in T9 of A2-wt by ¹H $R_{1\rho}$ RD

experiments. **d** Kinetic parameters derived from ¹H $R_{1\rho}$ RD experiments for T9 ᴺH3 in A2-P16 and A2-c⁷A16, as well as aromatic protons (ᶜH6 and ᶜH6) of the modified P16 base. Comparison of $\Delta\delta$ (ppm) with A2-wt confirms the presence of both HG (green) and ES2 (red) in A2-P16. Notably, the population of ES2 and the associated exchange rate $k_{ex(HG–ES2)}$, (orange) are elevated in A2-P16, while $k_{ex(WCF–HG)}$, (light blue) remains unchanged. In A2-c⁷A16, a HG-like state (light orange) with respective exchange rates (grey and black) is observed despite prior assumption[24] that steric hindrance between the A16 ᶜH7 and T9 ᴺH3 would prevent such a conformation. The error bars were estimated form Monte Carlo sampling of 500 replicas, representing mean ± 1 SD. Source data are provided as a Source data file.

canonical GpC binding site for ActD[44], CSP analysis suggests the presence of two binding modes (A and B) at 1:1 concentration (1.0 mм of each ActD and A2 DNA) and 298 K (Fig. 5a and 5b). These binding sites are localised around T9, G10 and G11, resembling various noncanonical binding sites such as the GpG and T(G)ₙT sites reported in the literature[45–51]. Overlaying the chemical shift data of ES2 in A2 wt onto these CSPs indicates that in binding mode B, the observed chemical shifts of ES2 for T9 ᴺH3 and A16 ᶜH2 align most closely (Fig. 5b and Supplementary Fig. 9), suggesting that this state is stabilised or promoted, indicating a potential role of ES2 in drug interactions.

**Structural models for ES2**
Molecular dynamics (MD) simulations were performed to investigate the WCF and HG conformations of the T9-A16 base-pair in A2-wt, A2-P16 and A2-c⁷A16 DNA constructs. The WCF conformation was modelled based on a reported NMR structure (PDBid: 5UZD)[30], while the HG conformation was generated by rotating the χ-dihedral angle of A16 by 180° followed by energy minimisation. Parameters for the modified nucleotides P16 and c⁷A were derived using standard two-stage restrained electrostatic potential fitting of $N^9$-methylated nucleobases with geometries optimised at the HF/6-31 G* level of theory[52,53].

These charges were combined with existing parameters for deoxyribose and phosphate groups of dA nucleotide in the Amber-OL15 force field[52] (see Supplementary File for details).

Each system was simulated for 200 ns, during which the structures remained stable, including those containing modified nucleotides (Supplementary Fig. 10). To estimate chemical shifts, 50 random frames were extracted between 2 and 200 ns, and a 3.3 Å radius around T9, which includes the base-pairing partner A16 as well as nucleotides from the base-pairs immediately above and below, was used for fragment generation with AFNMR[54] followed by GIAO-DFT calculation with Orca[55,56]. The efficacy of the AFNMR method compared to other empirical methods to predict the chemical shift of DNA was reported previously[28,54,57]. The relative chemical shift ($\Delta\delta$, ppm) between HG and WCF in A2 wt was reliably estimated for non-exchangeable protons (A16 ᶜH8, A16 ᶜH2, and A17 ᶜH2). However, as expected, for exchangeable imino protons (T9 ᴺH3, and T8 ᴺH3), higher variability was observed (Supplementary Fig. 11a). Nevertheless, the mean $\Delta\delta$ trends were consistent with the experimental observations. In A2-P16, similar trends in ¹H chemical shifts were observed, except for P16 ᶜH6, where the predicted $\Delta\delta$ differed in sign from the experimental value (Supplementary Fig. 11b). For A2-c⁷A16, the predicted $\Delta\delta$ for T9 ᴺH3

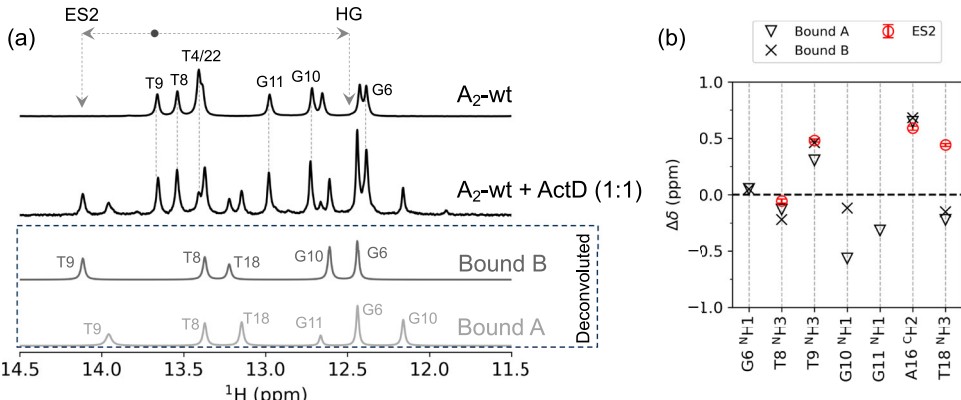

**Fig. 5 | Drug interaction and structural models of ES2. a** $^1$H imino spectra of free and Actinomycin D-bound $A_2$ DNA (1:1 molar ratio) at 298 K and pH 6.5 reveal two distinct bound states: Bound A (light grey) and Bound B (dark grey). Deconvoluted spectra for each bound state are shown below. Arrows indicate the chemical shifts corresponding to the ES2 and HG conformations for T9 $^N$H3 (black dot) in $A_2$ wt. Notably, the T9 $^N$H3 resonance in Bound B closely matches the ES2 position. **b** $\Delta\delta$ between the ActD-bound (triangles: Bound A; crosses: Bound B) and free $A_2$ DNA are overlaid with the experimental $\Delta\delta$ associated with ES2 (red). The alignment of chemical shifts for T9 $^N$H3, T8 $^N$H3, and A16 $^C$H2 between Bound B and ES2 supports the hypothesis that ES2 may contribute to drug binding. The error bars for the $\Delta\delta$ (ppm) in ES2 is derived from Monte Carlo sampling of 500 replicas. Source data are provided as a Source data file.

matched the experimental trend (Supplementary Fig. 11c), supporting the hypothesis that the minor conformation observed in $R_{1\rho}$ experiments (Fig. 4d and Supplementary Fig. 11b) represents an HG-like state.

To further explore the structural basis of ES2 implicated in the WCF−HG exchange, enhanced sampling via well-tempered parallel biased metadynamics[36–38,58,59] was employed. This method enables the exploration of transient intermediates along the transition pathway[60–63]. Previous simulations identified two primary mechanisms for the purine *anti*-to-*syn* flip: an intrahelical pathway with a small base opening angle and an extrahelical route with a larger opening angle[3,63]. However, these simulations have primarily focused on $A_6$-DNA with six contiguous A−T base-pairs and assumed a two-state exchange model with no experimental indication of an additional excited state.

To investigate ES2 in $A_2$ DNA (Fig. 1a and Fig. 3), the orientation of the A16 nucleobase was steered using five collective variables (CVs) (Supplementary Table 10, for a detailed description, see supporting information). The reweighted 2D free energy surfaces (FES) revealed two expected minima corresponding to WCF and HG conformations, as well as two additional minima potentially corresponding to ES2 (Fig. 6a). Structural clustering based on energy-weighted trajectories yielded representative conformers, from which chemical shifts were calculated (Supplementary Fig. 11d). The homogeneity of the cluster members in the chemical shift space and the convergence of the metadynamics simulation is shown in Supplementary Fig. 12. Agglomerative clustering[64] based on chemical shifts of all H, C and N atoms in T8, T9, G10, C15, A16 and A17 identified four major clusters (Fig. 6a and Supplementary Fig. 13). Two chemical shift clusters mapped precisely onto the known WCF and HG conformations, while two intermediate states, Model 1 and Model 2, were located along the WCF−HG transition pathway.

Calculated $\Delta\delta$ values for WCF/HG, WCF/model 1, and WCF/model 2 were of similar magnitude to experimental data for non-exchangeable A16 $^C$H8 and A16 $^C$H2 protons (Fig. 6b). This supports the possibility that both model 1 and model 2 represent conformers of ES2. In model 1, A16 adopts a conformation where the amino group forms a hydrogen bond with T9 O2, while in model 2, A16 is partially displaced from the helical axis (Fig. 6c). In $A_2$-P16, where the amino group of A16 is absent- increased exchange rate of WCF−ES2 was observed (Fig. 4d). Additionally, model 2 is observed to be consistently in a high-energy region of the 2D FESs for different CVs (Supplementary Fig. 13). These observations suggest that disruption of the A16-T9 hydrogen bond between A16 $^N$H6 and T9 O4/2, as in model 1, may be

the rate-limiting step, and is the more likely structural representation of ES2.

Overall, this study provides a detailed theoretical description of the influence of dipolar-coupled protons $^1$H$_{dip}$ in proximity to protons undergoing exchange during a $^1$H $R_{1\rho}$ RD experiment. We demonstrate that for $^1$H$_{dip}$ distances greater than 3 Å, as are the most common in typical DNA and RNA structures, with $\Delta\omega_{dip}$ either within or outside the screened offset range in an off-resonance experiment, the impact of cross-relaxation on $R_{1\rho}$ is negligible. This validates the use of standard exchange models to fit the relaxation dispersion data. Several effects were studied for cases when $^1$H$_{dip}$ is present at distances less than 3 Å and $\Delta\omega_{dip}$ falls within the screened offset range. These findings broaden the applicability of $^1$H $R_{1\rho}$ RD experiment, increasing the number of sites that can be probed for dynamics in nucleic acids.

Using $A_2$ DNA, we detected a previously underappreciated second excited state (ES2) in the WCF−HG transition. MD and metadynamics simulations were applied to reveal that ES2 likely corresponds to an intermediate state featuring a hydrogen bond between the A16 amino group and T9 O2 during the WCF−HG exchange pathway. Further-more, the increased exchange rate to ES2 observed in nebularine-modified $A_2$-P16 DNA, lacking the A16 amino group, supports a mechanistic role for this interaction in modulating the transition. Detecting the ES2 and HG conformers in both $A_2$-P16- and $A_2$ c$^7$A16-modified DNA, highlights the improved sensitivity of $^1$H $R_{1\rho}$ RD experiments compared to $^{13}$C and $^{15}$N $R_{1\rho}$ techniques. The ES2 con-formation was observed to be stabilised when $A_2$ DNA was combined with Actinomycin D, suggesting its potential role in binding anticancer drugs.

The ability to characterise transient, additional states like ES2 in the WCF−HG dynamics through integrated experimental and compu-tational approaches opens avenues for studying conformational landscapes of nucleic acids and their roles in molecular recognition.

## Methods
### Solid phase synthesis of modified DNA
5′-*O*-DMT-protected 3′-β-cyanoethyl phosphoramidites of dA, dC, dG, dT, dP (2′-deoxynebularine) and c$^7$dA, as well as the 5′-*O*-MMT-pro-tected 3′-β-cyanoethyl m$^1$dA phosphoramidite, were purchased from Glen Research, while other reagents were obtained from Sigma Aldrich.

Solid-phase synthesis of DNA oligonucleotides was carried out using standard phosphoramidite chemistry on a 1μmol scale with

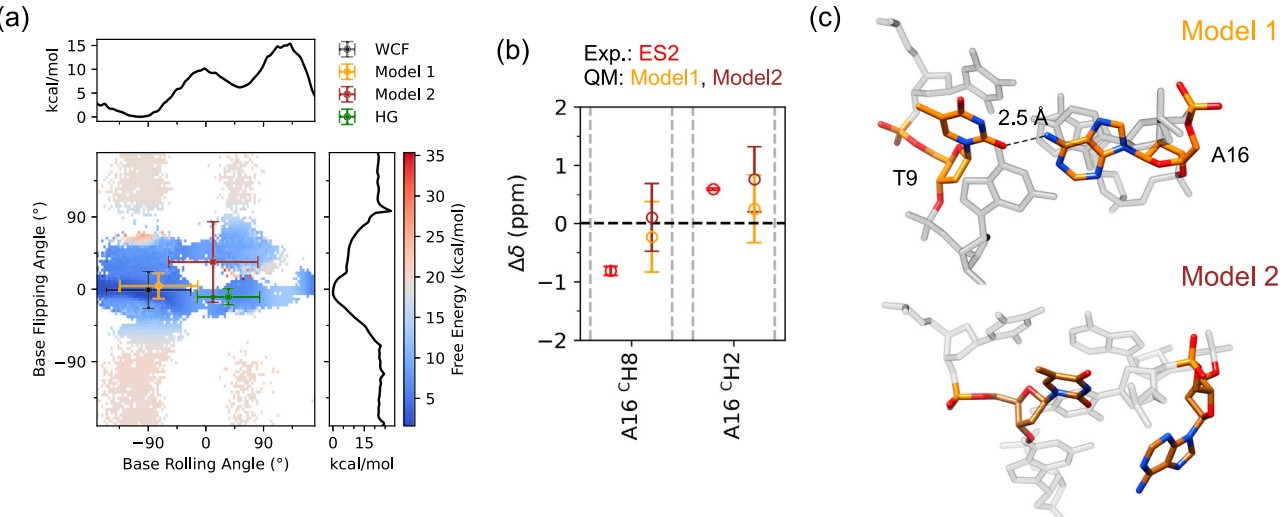

**Fig. 6 | Structural model of ES2. a** Reweighted 2D free energy surface (FES) from metadynamics simulations plotted as a function of base-flipping and base-rolling angle (χ-dihedral). Cluster centroids derived from chemical shift-based agglomerative clustering are overlaid: WCF (black) and HG (green) correspond to the expected minima. Model 1 (orange) and Model 2 (brown) map clusters in distinct regions, where Model 1 aligns with a local low-energy region, while Model 2 occupies a higher-energy region. This trend is consistent across multiple collective variables CV projections (Supplementary Fig. 12), suggesting Model 1 as a more plausible structural candidate for ES2. Error bars on the chemical shift-based cluster represent mean ± 1 SD among the cluster members (N for different clusters, WCF = 398, HG = 46, Model 1 = 105, Model 2 = 23) along the CVs. It is observed that the chemical shift clusters for WCF (black), HG (green) and Model 1 (orange) are structurally homogenous (Supplementary Fig. 12a). For Model 2, the error bars are larger and show some structural inhomogeneity, however most of the structures are localised in its unique high energy local minimum. One-dimensional projections of the FES are shown along the top and right axes. **b** Comparison of $\Delta\delta$ values, relative to WCF geometry, for non-exchangeable A16 $^{C}$H8 and A16 $^{C}$H2 aromatic protons in ES2 models (Model 1 (orange), and Model 2 (brown)) calculated using DFT with the corresponding experimental value of ES2 (red). Error bars represent mean ±1 SD across cluster frames. Both models exhibit chemical shift values consistent with the experimental ES2. **c** Representative minimum-energy structures from the Model 1 and Model 2 clusters. Model 1 features a hydrogen bond between the A16 amino group and T9 O2 (dashed line with heavy atom distance of 2.5 Å), while Model 2 exhibits partial flipping of A16 out of the helical axis. Source data are provided as a Source data file.

5-(ethylthio)-1$H$-tetrazole (ETT) as the activator. Phosphoramidites were employed as 50 or 70 mM (standard or modified DNA nucleotides, respectively) solutions in anhydrous MeCN.

Cleavage from the solid support and removal of base-labile protecting groups was achieved by treatment with a 1:1 mixture of 25% aq. $NH_3$ and 40% aq. $MeNH_2$ (1.0 mL) at ambient temperature for 30 min, after which the support was washed with an additional 0.5 mL of the same mixture. The combined solutions were heated to 65 °C for 30 min.

For m[1]dA-modified DNA, $N^6$-phenoxyacetyl and $N^2$-(4-isopropyl) phenoxyacetyl protecting groups were employed on the dA and dG phosphoramidites, respectively. Capping was performed with a combination of 5% phenoxyacetic anhydride in THF/pyridine and 16% 1-methylimidazole in THF and 20 mM $I_2$ in THF/$H_2O$/pyridine was used as the oxidiser[65]. After drying under reduced pressure, the solid support was treated with 2 M $NH_3$ in MeOH (1.0 mL) at ambient temperature for 24 h. It was washed with an additional 0.5 mL of the same solution; the combined solutions were evaporated to dryness, and the residue was dissolved in 1.5 mL of $H_2O$.

The crude solid-phase synthesis products were purified using Glen-Pak DNA purification cartridges according to the manufacturer's instructions. m[1]dA-modified DNA was eluted from the cartridge with 0.5% $NEt_3$ in MeCN/$H_2O$. Oligonucleotides were analysed by 20% denaturing PAGE. Yields were determined by UV absorbance and are reported in Supplementary Table 2.

**NMR sample preparation**
Both strands of the unmodified $A_2$ DNA ($A_2$-wt) duplex were purchased from Integrated DNA Technology (IDT) as standard desalting. The strands were mixed to a final concentration of 1.5 mM in 500 μL NMR buffer (15 mM NaP$_i$ pH 6.5, 25 mM NaCl, 0.1 mM EDTA). This solution was subjected to slow annealing during which it was heated to 95 °C for

5 min, followed by incubation at 65 °C, 37 °C, 25 °C and finally 4 °C for 30 min each. For modified $A_2$ DNA ($A_2$-P16, $A_2$-c[7]A16 and $A_2$-m[1]A16), the purified samples were buffer-exchanged to NMR buffer after annealing. The final samples were concentrated to the respective concentrations reported in Supplementary Table 3 in a final volume of 250 μL.

Actinomycin D (ActD) was purchased from Sigma Aldrich (A9415-2MG) and used without any further purification. Two milligrams of powder was dissolved in 160 μL of NMR buffer + 840 μL of MeOH giving a stock solution of 1.6 mM. 156 μL of this solution was evaporated under reduced pressure and the remaining solid was resuspended in 250 μL of 1 mM unmodified $A_2$ DNA in NMR buffer giving a 1:1 $A_2$ DNA:ActD solution. This solution was spiked with 6% $D_2O$ and transferred to in a 5 mm Shigemi tube for NMR measurements.

**NMR measurements and processing**
NMR measurements were performed on a 600.16 MHz Avance III-HD Bruker spectrometer equipped with a QCI-P cryoprobe. SOFAST HMQC[66,67] spectra were acquired with 128 increments and recorded with 64 scans each. The carrier frequencies of [1]H and [13]C were placed at 7.8 ppm and 142 ppm, respectively. Imino NOESY experiments were performed with a mixing time of 180 ms using the pulse sequence implemented in NMRlib[68]. The [1]H carrier frequency was set to 13 ppm, and the excitation bandwidth was 4 ppm. [1]H $R_{1\rho}$ RD experiments were performed using previously reported pulse sequences[14,69]. The acquisition parameters are tabulated in Supplementary Table 6. NMR spectra were processed using Bruker Topspin 3.6.3. Deconvoluted peak intensities obtained using "mdcon" and the corresponding signal-to-noise ratios calculated with "sino" in Topspin were exported to plain text for further data analysis. Temperature-dependent $R_{1\rho}$ experiments were performed in steps of 5 K from 283 to 303 K for wild type $A_2$ DNA. [1]H $R_{1\rho}$ RD experiments on modified $A_2$ DNA duplexes were performed

at 278 K. All other spectra were recorded at 298 K with an inter-scan delay of 1.5 s unless indicated otherwise.

### $R_{1\rho}$ data fitting

Peak intensities ($I$) as a function of spinlock duration $\tau_{SL}$ were fitted to a mono-exponential decay to obtain the underlying rotating frame relaxation rates $R_{1\rho}$. Data points were weighted according to their uncertainties given by the root mean square of the baseline noise of each spectrum, which is defined as rms = $I/(2 \times \text{sino})$. Confidence intervals for the resulting $R_{1\rho}$ values were estimated from the standard deviation obtained by Monte Carlo resampling of the original datasets with 500 replicas.

For analysis of conformational exchange, $R_{1\rho}$ was fitted as a function of spinlock strength $\omega_{SL}$ and offset $\Omega_{SL}$ with weights given by the standard deviation. Exchange processes were modelled as two-state or three-state without or with minor state exchange by taking $R_{1\rho}$ as the least negative real eigenvalue of the corresponding Bloch-McConnel matrix (see below for a full description of the matrices)[32]. Models were ranked using the Akaike information criterion with small sample size correction (AICc), Bayesian information criterion (BIC, difference ≥10) and *F*-test (95% confidence level)[6]. The goodness of fit was further judged based on the distribution of fit residuals, which are ideally random and centred around $y = 0$ for a model that captures the exchange process appropriately. Confidence intervals for each exchange parameter of the best model were estimated from the respective standard deviation obtained by Monte Carlo resampling of the original datasets with 500 replicas.

All data analysis was performed using a custom programme written in Python 3.10. Linear algebra operations for obtaining exact numerical eigenvalues were implemented with the NumPy package. Non-linear least-squares minimisation with the Levenberg-Marquardt algorithm was performed using the lmfit package[70]. The attached Supplementary Data 1, provides the data for the individual and global fits for the protons subjected to $^1H$ $R_{1\rho}$ RD experiments. The AICc and BIC criteria are reported for each of the three-state models. The best model from these statistical tests and the F-test is highlighted with thick border.

### Molecular dynamics (MD) simulation

MD simulations of $A_2$ DNA were performed for both the WCF and HG conformations of the nucleobase A16. The lowest energy structure from PDBid: 5UZD[30] was used as a model for the WCF conformation. For the HG conformation the glycosidic χ dihedral angle of A16 was rotated by 180°. The Amber14SB force field with OL15 parameters for DNA (https://fch.upol.cz/ff_ol/gromacs.php) was used to simulate both the models using the GROMACS 2023.4 suite. All-atom MD simulations were performed for 200 ns using a TIP3P water model in a dodecahedral box. The system was first neutralised using of $Na^+$ ions and the final NaCl concentration was set to 25 mM. The system was energy minimised via the steepest decent gradient method to a maximum force constant <1000 kJ/mol, equilibrated for a total of 200 ps at 300 K using a velocity rescale scheme and 1 bar pressure using the Parrinello-Rahman barostat. A Verlet nonbonded cut-off scheme with grid neighbour search and 1.0 nm cut-off for van der Waals interaction with energy and pressure dispersion correction was used. Particle Mesh Ewald with fourth order cubic interpolation was used for Coulomb interactions. All bonds were constrained using the LINCS algorithm. During the MD run, a 2 fs integration step was used and the trajectory was saved every 10 ps. The trajectory was analysed using Plumed 2.7[36,59], UCSF Chimera 1.17.3 and Python 3.11. Simulations were performed in the Tetralith HPC cluster via the National Academic Infrastructure for supercomputing in Sweden (NAISS). PDB models and molecular dynamics parameters files are provided in Zenodo (https://doi.org/10.5281/zenodo.17155220).

### Parameterisation of modified nucleotides for MD simulation

To derive force field parameters for deoxyribonucleotides of purine (P, residue code PRN) and 7-deazaadenine ($c^7A$, residue code 7DA), geometries of the respective $N^9$-methylated nucleobases were first optimised in the gas phase with tight convergence criteria and characterised as stationary points on the potential energy surface by analytical vibrational frequency calculation using Orca 5.0.4[55,56]. The nucleobases were subjected to two-stage restrained electrostatic potential (RESP) fitting with default parameters and standard hyperbolic restraints[52,53] using Psi4 1.9.1[71,72]. For this, the charge of the capping exocyclic methyl group was set to +0.1053 and 20 equally weighted, randomised orientations of the nucleobases were considered. GROMACS-compatible.itp files were generated with ACPYPE[73] (options -c user -a amber -o gmx) and combined with deoxyribose and phosphate parameters for unmodified adenine deoxyribonucleotide (residue code DA) as defined in the Amber force field.

### Metadynamics simulation

GROMACS 2021.3 patched with PLUMED 2.7.2 was used to perform a parallel biased well-tempered metadynamics simulation by biasing five collective variables (Supplementary Table 11). The simulation was performed for 200 ns until convergence with the height of the gaussians set to 0.5 kJ/mol deposited every 500 steps. The sigma of the gaussian height was set to the ½ of the standard deviation obtained from the unbiased simulations. The 1D free-energy surface (FES) for each collective variable (CV) were obtained from the reweighted bias energy while the 2D FES were obtained by using binned_statistic_2d from scipy.stats module of scipy using each combination of CV and the reweighted bias energy (Supplementary Fig. 12). All the files used for the simulations and analysis are provided in the Zenodo repository (https://doi.org/10.5281/zenodo.19420109).

### Chemical shift calculation

Calculation of chemical shifts for T8, T9, G10, C15, A16 and A17 from MD frames was done with the help of an automated fragment generation approach[54]. Before compiling version 1.8 of the AFNMR programme, the value of 'MAXNRES' in afnmr-F90 was updated to include the modified nucleotides, and their residue codes were added in the form of 'nresn(34) = 'PRN' etc. The compiled programme was checked using the built-in test suite. AFNMR requires Amber-compatible frcmod and.lib files for modified nucleotides, which were generated using ACPYPE (options -c user -a amber -o gmx) from.mol2 files of the respective residues with charges as derived during MD parametrization.

AFNMR was run on each frame with the following options:
*-list "{{8..10},{15..17}}" -orca -nomin -mixedb -frcmod ${MOD_NAME}_AC.frcmod -offlib ${MOD_NAME}_AC.lib -workdir.*

This produces residue-centric fragments that include all neighbouring residues, water molecules and ions in a sphere of 3.3 Å, i.e. the direct base pairing partner plus residues from the base pair above and below. The rest of the system is projected onto the fragment as a set of point charges. Chemical shielding tensors were calculated in Orca 5.0.4[55,56] by GIAO-DFT with the OLYP functional using a pcSseg-2 basis set on the central residue, pcSseg-1 basis set on all other atoms[74] and def2/J auxiliary basis set[75] for the RI-J approximation. Chemical shifts were obtained using the built-in referencing routine of AFNMR, but were not corrected further as only relative chemical shift changes between MD frames/conformations were of interest.

For analysis of chemical shift changes from conventional MD trajectories of $A_2$-wt, $A_2$-P16, and $A_2$-$c^7A16$ with nucleotide 16 either in the WCF or in the HG conformation, chemical shifts of each atom type were averaged over all 50 frames and standard deviations were calculated.

## Structural and chemical shift clustering

(a) *Benchmarking AFNMR using unbiased simulation*: To validate the accuracy of the AFNMR chemical shift predictions, chemical shifts were first computed from an unbiased molecular dynamics simulation. Fifty frames were randomly selected from the unbiased trajectory for DFT-based chemical shift calculations. The first 2 ns of the trajectory were discarded because the backbone RMSD undergoes significant changes during this period (Supplementary Fig. 10), indicating equilibration of the system. The remaining frames represent equilibrated ensembles corresponding to either WCF or HG base-paired conformations. As shown in Supplementary Fig. 11a–c, the AFNMR-predicted chemical shifts agree well with the experimentally measured values and previously reported literature data for both WCF and HG states. The agreement is within the typical uncertainty of DFT-based chemical shift calculations ($^1$H (non-labile) = 0.2–0.3 ppm, $^1$H (labile) ∼ 1 ppm, $^{13}$C = 2–3 ppm and $^{15}$N = 4–5 ppm[54,76–78]). This benchmarking confirms that the AFNMR protocol provides reliable chemical shift predictions for the conformational states sampled in our simulations.

(b) *Metadynamics trajectory analysis and DFT calculation*: To characterise the structural ensembles sampled during metadynamics, a three-dimensional (3D) structural clustering analysis was performed using the GROMOS algorithm implemented in GROMACS with a backbone RMSD cutoff of 0.2 nm. This procedure yielded 123 clusters containing at least three members each (deposited in https://doi.org/10.5281/zenodo.17155220). Representative structures from these clusters were subsequently used for DFT-based chemical shift calculations.

i. *Homogeneity assessment of the cluster in the chemical shift space*: To evaluate whether the structures within each cluster exhibit similar NMR properties, DFT calculations were performed on approximately one-third of the members within each cluster. While for the clusters with ≤5 members all the frames were used. For each structure, the chemical shift deviation ($\Delta\delta$) relative to the cluster average was computed (Supplementary Fig. 12a). Most of the deviations fall within the typical DFT uncertainty range, indicating that the chemical shift variation among cluster members is small. The distribution of $\Delta\delta$ values is centred near zero, demonstrating that each structural cluster is also homogeneous in chemical shift space. This result justifies using the cluster centre as a representative structure for subsequent chemical shift analysis.

ii. *Clustering of chemical shift*: To identify distinct ensemble states based on NMR observables, the chemical shifts computed for the representative structure of each 3D cluster were subjected to agglomerative clustering using a Euclidean distance metric with Ward linkage. Because each structural cluster is homogeneous in chemical shift space, the cluster centre adequately represents the chemical shift properties of that cluster.

The clustering analysis included all calculated $^1$H, $^{13}$C, and $^{15}$N chemical shifts of the six central residues (T8, T9, G10, C15, A16, and A17). Different numbers of output clusters were tested to determine the number that best recovers distinct structural states from the chemical shift data. Four output clusters provided the most meaningful separation. When fewer clusters were used, chemically and structurally distinct states became merged; for example, clustering into three groups fails to distinguish the HG and Model 1 geometries. Mapping the resulting chemical shift clusters back onto the five collective variables used in the metadynamics simulations reveals four distinct conformational ensembles corresponding to WCF, HG, Model 1, and Model 2 (Fig. 5c and Supplementary Fig. 11).

Projecting the agglomerative clustering results onto the original 3D structural clusters shows that multiple first round structural clusters map onto the same chemical shift–based ensemble state. This is expected because the initial structural clustering was performed using the backbone of the entire DNA duplex, whereas the chemical shift clustering focuses only on the six central residues. Structural variations in other regions of the DNA therefore collapse into a smaller number of chemically distinct conformers when analysed in the chemical shift space. After assigning each structural cluster to one of the four conformational ensembles, the stability of these states was further evaluated to assess the convergence of the metadynamics simulations.

iii. *Stability assessment of the ensemble states*: To quantify the stability of the sampled ensembles, the unbiased population of each structural cluster was computed from the reweighted metadynamics trajectory. The population of a cluster was calculated as the sum of the statistical weights of all frames belonging to that cluster, normalised by the total weight of the trajectory. The statistical weights were obtained using the Tiwary–Parrinello reweighting method[79], which accounts for the metadynamics bias potential. Statistical uncertainties in the populations were estimated using block averaging of the reweighted trajectory. The resulting cluster populations exhibit small statistical uncertainties (Supplementary Fig. 12b, c), indicating that the estimated probabilities of the structural states are stable. This analysis demonstrates that the metadynamics simulation provides a reliable estimate of the equilibrium distribution across the sampled conformational ensembles and supports the convergence of the simulation in terms of structural state populations.

iv. *Comparison of clustered chemical shifts to experimental data*: The predicted chemical shifts for the ensembles corresponding to Model 1 and Model 2 were compared with experimentally observed chemical shifts associated with the ES2 state. Specifically, the following resonances were analysed: T9 $^N$H3, T8 $^N$H3, A16 $^C$H8, A16 $^C$H2, and A17 $^C$H2 in A$_2$ wt; P16 T9 $^N$H3 and P16 $^C$H8 in A$_2$ P16; and c$^7$A T9 $^N$H3 in A$_2$ c$^7$A (Supplementary Fig. 11a, b and Fig. 5d).

Considering the typical uncertainty of DFT-based chemical shift calculations together with the standard deviation within each ensemble, both Model 1 and Model 2 reproduce the experimentally observed chemical shift values for ES2. However, comparison with the experimental data obtained from modified DNA constructs indicates that Model 1 provides a better overall agreement (see main text for details).

Consistent with this interpretation, the population and free energy analysis (Supplementary Fig. 12c) shows that both Model 1 and Model 2 correspond to low-population, high-energy structural ensembles relative to the dominant WCF and HG states, as expected for an excited-state conformational ensemble.

## Reporting summary

Further information on research design is available in the Nature Portfolio Reporting Summary linked to this article.

## Data availability

All data generated in this study are provided in Supplementary information, source data files, supplementary data 1 file. Raw NMR data, additional source data related to $R_{1\rho}$ relaxation dispersion experiments, molecular dynamics trajectory and data of quantum calculation to obtain NMR chemical shift is deposited in Zenodo repository which can be accessed from https://doi.org/10.5281/zenodo.17155220. PDB codes of previously published structures used in this study are 5UZD, 5UZI, and 1EKA. Source data are provided with this paper.

## Code availability

Code to simulate NMR parameters is available is deposited in Zenodo repository which can be accessed from https://doi.org/10.5281/zenodo.19420109.

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

## Acknowledgements

R.D. acknowledges funding from European Union's Horizon 2020 research and innovation programme under the Marie Skłodowska-Curie Action (MSCA) [101067627, project: ECONOMICS]; C.S. acknowledges funding by an EMBO postdoctoral fellowship (ALTF 1011-2020); K.P. acknowledges funding from Wallenberg Academy Fellow (KAW 2019, 0227), project grant from the Knut och Alice Wallenberg foundation (KAW 2016.0087), Cancerfonden (CAN 2018/715 and 21 1770 Pj-BF 1), KI consolidator grant (2-2111/2019) and Karolinska Institute for the help with the purchase of our 600 MHz NMR.

## Author contributions

R.D. and C.S. performed most of the experiments, conceptualisation, formal analysis. C.S. performed most of the visualisation. J.I. implemented NMR methods and edited the manuscript. R.D., C.S. and K.P. acquired funding. KP supervised the study and provided project administration and analysis. R.D., C.S. and K.P. contributed equally to writing, review & editing.

## Funding

## Competing interests

The authors declare no competing interests.
