## [Transparent Peer Review file · Nature Communications]

^1H R1 ρ Relaxation Identifies a Hidden Intermediate in DNA Base-Pairing

Corresponding Author: Professor Katja Petzold

Version 0:

Reviewer comments:

Reviewer #1

(Remarks to the Author)

This manuscript, titled " ^1H R1 ρ Relaxation Identifies a Hidden Intermediate in DNA Base-Pairing," is authored by Rubin Dasgupta, Christian Steinmetzger, Julian Ilgen, and Katja Petzold. The study utilizes ^1H R1 ρ relaxation dispersion NMR spectroscopy to investigate the conformational dynamics of DNA base-pairing. The authors first establish a reliable methodology by quantifying cross-relaxation artifacts, demonstrating their negligible effect for inter-proton distances greater than 3 Å. Applying this method to an A2 DNA duplex, they discover a previously unknown second excited state (ES2), which acts as an intermediate in the well-established Watson-Crick to Hoogsteen equilibrium. A structural model for ES2 is proposed based on a combination of NMR data, DNA modifications, metadynamics simulations, and DFT chemical shift calculations. Furthermore, the study provides experimental evidence that the anticancer drug Actinomycin D stabilizes this ES2 intermediate, suggesting a novel mechanism for drug-DNA interaction. The work presents both a methodological advancement for NMR studies and a mechanistic discovery of a drug-stabilized intermediate in DNA dynamics.

While the manuscript addresses an important topic in DNA dynamics, its suitability for a broad-readership journal like Nature Communications is questionable for the following reasons:

The manuscript is exceptionally specialized. The theoretical discussions, particularly regarding the quantification of cross-relaxation artifacts in ^1H R1 ρ relaxation dispersion NMR experiments, are highly technical. This content is likely accessible only to experts in NMR spectroscopy, making it difficult for the general readership of Nature Communications to fully grasp the significance of the methodological advancements. Furthermore, a review of the references reveals that not a single article from Nature Communications has been cited. This suggests that the work may not be well-aligned with the topics and discussions that are currently of interest to the journal's primary audience. Given these points, the manuscript would be better suited for a specialized journal focusing on NMR methodology, structural biology, or biophysical chemistry. Submitting to such a venue would ensure the paper reaches an audience that can fully appreciate its technical depth and scientific contribution.

Additionally, the following points should be addressed for revision:

The assertion that cross-relaxation artifacts are negligible beyond 3 Å is an oversimplification, as these effects are fundamentally governed by the spectral density function, $J(\omega)$, not just distance. The model used here for $J(\omega)$ is itself flawed, as it assumes a rigid sphere model, which is an inaccurate approximation for a cylindrical molecule like DNA. A more rigorous analysis accounting for the anisotropic tumbling of DNA has already been reported by Furukawa et al. (Nucleic Acids Research 2020, 49 (2), 1152). The authors should therefore re-evaluate and discuss their claim in the context of this more appropriate theoretical framework.

The Introduction section contains some of the study's results. This disrupts the paper's logical structure. These results should be moved to the appropriate section, allowing the Introduction to focus solely on establishing the scientific background and research objectives.

The manuscript frequently uses the term " ^1H R1 ρ " as a shorthand for " ^1H R1 ρ relaxation dispersion". These terms are not interchangeable. A ^1H R1 ρ measurement yields a single relaxation rate, whereas relaxation dispersion is the experiment measuring these rates as a function of the spin-lock field. The authors should revise the text to use the full, correct terminology to ensure clarity and technical precision.

For improved clarity and adherence to scientific typesetting conventions, the formatting of variables should be revised. Symbols representing physical quantities should be consistently italicized. Conversely, non-variable subscripts and superscripts that serve as descriptive labels should be set in Roman (upright) type. A thorough review is recommended to ensure this formatting standard is applied consistently throughout the main manuscript and all accompanying Supporting Information.

The units s⁻¹ and Hz are used inconsistently for rates and frequencies throughout the manuscript. For clarity, the authors should adopt a single, consistent convention and apply it uniformly across the text, figures, tables, and Supporting Information.

On page 2, line 44, the abbreviation "ES" is used as a plural noun ("ES are..."), which is inconsistent with its singular definition ("a higher energy, excited state (ES)") on line 41. This could create ambiguity for the reader.

In Figure 1, the structural representation in sub-panel (f) appears to be drawn at a larger scale compared to those in sub-panels (d), (e), and (g).

In the Figure 1 caption, the description "The position of the cross-peaks (in Hz)" is ambiguous. It is not clear what this position is referenced against.

On page 4, line 114, the term "R1_p" should be correctly formatted with an italicized 'R' and a '1_p' subscript.

The manuscript inconsistently uses both uppercase (X, Y, Z) and lowercase (x, y, z) letters to denote Cartesian axes. For clarity and consistency, a single convention should be adopted and applied uniformly throughout the text and figures.

On page 9, line 237, the phrase "these interactions" is ambiguous. For improved clarity, please specify that this refers to the dipolar interactions between the aforementioned proton pairs.

The caption for Figure 3 should be expanded for clarity by defining "red. χ^2 " (reduced chi-squared). Additionally, the "Residuals" plot is presented without any explicit mention or discussion in the main text.

On page 12, line 304, the notation "NH₃/1" is used without definition. This specialized terminology is potentially ambiguous.

On page 12, line 306, the phrases "lower chemical shift" and "higher chemical shift" are imprecise. For improved technical accuracy and adherence to standard NMR conventions, it is strongly recommended to use the more specific terms "upfield shift" and "downfield shift", respectively. This terminology should be used consistently throughout the manuscript.

The symbol $\Delta\delta$ is used inconsistently. It appears in Figures 3 and 4 without definition, while the caption for Figure 5 provides two different definitions ("Chemical shift difference" and "Relative chemical shift"). For clarity, a single, explicit definition of $\Delta\delta$ should be provided at its first use and applied uniformly throughout the manuscript.

Reference 10: This citation is incomplete, as it is missing the final page number.

Reference 15: The citation to Angew. Chem. with a volume number of 4 and no page numbers for the year 2016 appears to be incorrect. This entry should be verified and corrected.

(Remarks on code availability)

I confirmed that the code is available for download, but I have not actually used or tested it.

Reviewer #2

(Remarks to the Author)

The manuscript by Dasgupta et al. presents a significant advancement in the study of DNA dynamics by addressing a key technical limitation of relaxation dispersion NMR. The authors first systematically quantify the effects of cross-relaxation artifacts, demonstrating that they are negligible for protons more than 3 Å apart, which validates the technique for broader use in nucleic acids. Applying this refined method to a model DNA duplex, they discover a previously hidden, transient intermediate state (ES2) that extends the well-known two-state equilibrium between Watson-Crick-Franklin and Hoogsteen base pairs. The study further characterizes this novel state through a combination of chemical modifications, metadynamics simulations, and DFT calculations to propose a structural model. Finally, the biological relevance of this discovery is highlighted by showing that the ES2 conformation is stabilized by the anticancer drug Actinomycin D, suggesting a new role for transient DNA structures in drug-binding interactions.

Our expertise is mainly in Molecular Dynamics (MD) simulations, enhanced sampling, and integration with experimental data, including NMR. Therefore, our review will mostly focus on the modeling part of the manuscript. Our overall assessment is that this manuscript presents a compelling story, composed by combining both experimental NMR data and molecular modeling. However, we have some concerns regarding the latter part that we invite the authors to address in a revised version of the manuscript.

Major concerns

1. Our understanding of the pipeline used by the authors to structurally characterize the ES2 state, starting from its experimental chemical shifts, is as follows:

- Enhanced-sampling metadynamics simulations are performed.
- The resulting trajectory is clustered based on conformational similarity to yield distinct structural groups (3D clusters).
- Chemical shifts (CS) are then calculated for a representative structure from each 3D cluster using Density Functional Theory (DFT).
- These calculated CS are subsequently used for a second clustering analysis, which reveals four major groups (CS clusters).
- One of these CS clusters, designated as Model 1, is identified as the preferred structural model for ES2.

Based on this (hopefully correct) understanding:

a) It is unclear whether the 3D clusters are homogeneous with respect to their predicted CS, i.e. whether members of the same cluster exhibit similar (predicted) CS values. While we acknowledge the computational expense of DFT calculations, the authors should demonstrate, for at least one or two representative clusters, that their members have indeed similar predicted CS. This validation is crucial, as the selection of only the cluster center for CS calculation and subsequent CS clustering may not be representative of the entire conformational ensemble.

b) Similarly, after clustering in the CS space, the authors should assess whether the resulting CS clusters are structurally homogeneous. A straightforward approach would be to project the centers of all 3D clusters onto the 2D free-energy surfaces in Figure S12, color-coding each point according to its assigned CS cluster. This visualization would make it immediately apparent whether members of the same CS cluster are structurally related or are instead scattered across disparate regions of the conformational space.

c) The ES2 excited state is expected to correspond to a local, high-energy minimum on the free-energy surface, likely represented by an ensemble of conformations in rapid exchange that gives rise to a population-averaged experimental CS. In light of the two preceding points, we find the strategy used by the authors for the structural characterization of ES2 not entirely convincing. In our view, a more rational approach would be to identify one or more conformational clusters located in local, high-energy minima from the MD simulations and then determine if these clusters, either individually or as an ensemble average, match the observed ES2 chemical shifts. This contrasts with the current method of clustering in CS space, which risks grouping structurally diverse models from different parts of the conformational landscape. One possible direction would be to exclude the 3D clusters corresponding to the canonical Watson-Crick-Franklin and Hoogsteen states and then apply a Maximum Entropy (MaxEnt) reweighting to the remaining clusters to find the ensemble that best reproduces the experimental ES2 data.

d) Finally, it is unclear why the authors opted for DFT calculations over empirical CS predictors. For proteins, it is generally understood that the accuracy of DFT-based predictors is not necessarily superior to that of empirical methods, such as SHIFTX2. The authors should comment on this choice and: i) report the typical error associated with DFT chemical shift calculations for DNA and compare it to the performance of empirical predictors; and ii) explicitly account for this prediction error when comparing calculated and experimental CS for ES2. We believe that looking for a specific conformation (or ensemble) that matches the experimental CS below the typical error in the CS computation is not a robust validation of the model.

2. Our second major concern relates to the methodology and validation of the enhanced-sampling simulations.

a) First, the manuscript lacks a rigorous assessment of simulation convergence, which is essential, particularly given the relatively short 200 ns simulation length. A standard validation would involve presenting time-series plots for each metadynamics CV alongside a block analysis to estimate the statistical error on the 1D or 2D free-energy surfaces. Leveraging the authors' existing 3D cluster analysis, an even more direct approach would be to report the reweighted, unbiased population of each cluster with its associated statistical error.

b) Second, the rationale for selecting the specific collective variables (CVs) is not completely clear. In particular, two of the chosen CVs (the 'Base Flipping' angle and the 'T9 O4–A16 N1 distance') do not effectively discriminate between the WCF and HG states, questioning their relevance as a CV in metadynamics simulations. Furthermore, Table S11 also reports as CV the 'Centre of Mass C15 – A17' (also with no difference between the WCF and HG states), without clarifying whether this variable was ultimately used in the simulations.

Minor concerns

1. In Figure S11, it is unclear what the dots and error bars of the CS calculated with DFT represent.
2. Coloring in Figure S11d is a bit confusing, as red and brown look very similar to us.
3. Figure S11d should report the typical error in DFT calculations of CS to appreciate whether the observed differences are significant given this typical error.
4. In the caption of Figure S12, it is unclear what an "Energy-weighted" free energy is.

(Remarks on code availability)

The repository contains sufficient data to reproduce the simulations reported in the paper. We appreciate the effort of the authors to share this data to the community.

Reviewer #3

(Remarks to the Author)

(Remarks on code availability)

Version 1:

Reviewer comments:

Reviewer #1

(Remarks to the Author)

The authors have adequately addressed all points raised in the initial review, with one minor exception regarding the reference list.

While Reference 15 was corrected, the identical paper is duplicated as Reference 29.

(Remarks on code availability)

Reviewer #2

(Remarks to the Author)

The authors addressed satisfactorily all my previous concerns and revised the manuscript accordingly.

I would like just to point out that in my original report I never stated that "DFT-based predictors are not superior to empirical methods for nucleic acids", as remarked by the authors in their rebuttal. Actually, I was pointing out that *for proteins*, it is generally understood that the accuracy of DFT-based predictors is not necessarily superior to that of empirical methods. With my statement, I was inviting the authors to make a comment regarding DFT vs empirical predictors in nucleic acids, as I was not familiar (and maybe the reader as well) with their performances for these molecules. In any case, the authors ultimately addressed this point by citing previous works where these forward models were extensively tested on nucleic acids.

(Remarks on code availability)

Reviewer #3

(Remarks to the Author)

(Remarks on code availability)

Reviewer #1

This manuscript, titled "1H R1ρ Relaxation Identifies a Hidden Intermediate in DNA Base-Pairing," is authored by Rubin Dasgupta, Christian Steinmetzger, Julian Ilgen, and Katja Petzold. The study utilizes 1H R1ρ relaxation dispersion NMR spectroscopy to investigate the conformational dynamics of DNA base-pairing. The authors first establish a reliable methodology by quantifying cross-relaxation artifacts, demonstrating their negligible effect for inter-proton distances greater than 3 Å. Applying this method to an A2 DNA duplex, they discover a previously unknown second excited state (ES2), which acts as an intermediate in the well-established Watson-Crick to Hoogsteen equilibrium. A structural model for ES2 is proposed based on a combination of NMR data, DNA modifications, metadynamics simulations, and DFT chemical shift calculations. Furthermore, the study provides experimental evidence that the anticancer drug Actinomycin D stabilizes this ES2 intermediate, suggesting a novel mechanism for drug-DNA interaction. The work presents both a methodological advancement for NMR studies and a mechanistic discovery of a drug-stabilized intermediate in DNA dynamics.

- We thank the reviewer for the thorough review.

While the manuscript addresses an important topic in DNA dynamics, its suitability for a broad-readership journal like Nature Communications is questionable for the following reasons:

The manuscript is exceptionally specialized. The theoretical discussions, particularly regarding the quantification of cross-relaxation artifacts in 1H R1ρ relaxation dispersion NMR experiments, are highly technical. This content is likely accessible only to experts in NMR spectroscopy, making it difficult for the general readership of Nature Communications to fully grasp the significance of the methodological advancements.

- The reviewer is correct that parts of the theoretical framework - particularly the quantitative treatment of cross-relaxation effects in 1H R1ρ relaxation dispersion experiments - are necessarily technical and will be most directly accessible to readers with expertise in NMR spectroscopy. However, this theoretical analysis represents only part one component of the manuscript. It is a necessary foundation that enables the experimental findings presented in the study; without rigorously defining the boundaries under which cross-relaxation artifacts can be excluded, the subsequent biological and mechanistic conclusions would not be justified.

Importantly, the manuscript does not present a methodological advance in isolation. Rather, it establishes clear, practically applicable conditions under which 1H R1ρ RD experiments can be reliably interpreted and then leverages this framework to uncover a previously hidden intermediate in DNA base-pairing dynamics, including its modulation by a small-molecule drug. The broader conceptual advance lies in expanding the applicability of an NMR experiment to nucleic acids at natural abundance while simultaneously revealing new mechanistic insight into DNA conformational landscapes.

To improve accessibility for a broad readership, we have revised the results section to provide a more intuitive, concept-driven description of the cross-relaxation artifact and its mitigation. The detailed mathematical derivations and extended simulations have been moved to the Supporting Information (p10-11), where interested specialists can find the full theoretical treatment. We believe this restructuring maintains scientific

rigor while enhancing clarity and readability for the general audience of *Nature Communications*.

Furthermore, a review of the references reveals that not a single article from Nature Communications has been cited. This suggests that the work may not be well-aligned with the topics and discussions that are currently of interest to the journal's primary audience. Given these points, the manuscript would be better suited for a specialized journal focusing on NMR methodology, structural biology, or biophysical chemistry. Submitting to such a venue would ensure the paper reaches an audience that can fully appreciate its technical depth and scientific contribution.

- The suitability of a manuscript for *Nature Communications* cannot be assessed based on whether references from that specific journal appear in the bibliography. The reference list contains a broad range of high-impact journals spanning nucleic acid biology, chemical biology, structural biology, biophysics, and physical chemistry, including multiple citations to *Nature*, *Nucleic Acids Research*, *Biochemistry*, *Journal of the American Chemical Society*, *Nature Chemical Biology*, *Nature Structural & Molecular Biology*, *Angewandte Chemie*, and *Biophysical Journal*. This diversity reflects the interdisciplinary nature of the work and the breadth of the intended audience. The manuscript is not confined to NMR methodology, as we have identified a previously hidden excited state in DNA and propose a potential biological function for this state. These findings address fundamental questions in nucleic acid structure–function relationships and genome chemistry, which are of direct relevance to a broad biological and chemical readership. Interestingly, the reviewer him/herself suggests that the manuscript would be appropriate for NMR, biophysics, or structural biology journals, which already acknowledges that the work spans multiple scientific communities. This breadth is fully consistent with the scope of *Nature Communications*, which publishes cross-disciplinary studies with broad conceptual impact. We therefore maintain that the manuscript is well suited for *Nature Communications*.

Additionally, the following points should be addressed for revision:

*The assertion that cross-relaxation artifacts are negligible beyond 3 Å is an oversimplification, as these effects are fundamentally governed by the spectral density function, $J(\omega)$, not just distance. The model used here for $J(\omega)$ is itself flawed, as it assumes a rigid sphere model, which is an inaccurate approximation for a cylindrical molecule like DNA. A more rigorous analysis accounting for the anisotropic tumbling of DNA has already been reported by Furukawa et al. (*Nucleic Acids Research* 2020, 49 (2), 1152). The authors should therefore re-evaluate and discuss their claim in the context of this more appropriate theoretical framework.*

- Thank you to the reviewer to pointing out that we did not clearly state the limitations of the model we used, which has now been rectified. Such a short DNA, as used here, is quite well approximated with a spherical model, where less parameters need to be fitted (Occam's Razor). To satisfy the reviewer, we have now included the calculations using the anisotropic definition of the spectral density function for an axially symmetric molecule such as the A₂ DNA used in this study in the supporting information with subheading of Spectral density function for axially symmetric molecule (a case of 12 base-pair A₂ DNA) in the Materials and Methods. Figure S1 was extended by subfigure S1i, highlighting the effect of including the anisotropic parameters do not alter the

conclusion, but rather strengthens the observation made using the isotropic spectral density function. Here, the observed distance limit is 2.8Å, above which the effect of cross-relaxation is minimal. With this observation we propose a conservative estimate of 3Å so that this study can be applied to wide variety of biomolecules.

The Introduction section contains some of the study's results. This disrupts the paper's logical structure. These results should be moved to the appropriate section, allowing the Introduction to focus solely on establishing the scientific background and research objectives.

It is common practice in many journals to conclude the introduction with a brief forward-looking summary of the study's objectives and principal findings in order to clearly position the work and guide the reader. The remaining introduction establishes the scientific background, outlines the existing limitations in the field, and defines the conceptual gap addressed by this study. After carefully re-examination, no results are being mentioned in the remaining introduction.

The manuscript frequently uses the term "1H R1ρ" as a shorthand for "1H R1ρ relaxation dispersion". These terms are not interchangeable. A 1H R1ρ measurement yields a single relaxation rate, whereas relaxation dispersion is the experiment measuring these rates as a function of the spin-lock field. The authors should revise the text to use the full, correct terminology to ensure clarity and technical precision. For improved clarity and adherence to scientific typesetting conventions, the formatting of variables should be revised. Symbols representing physical quantities should be consistently italicized. Conversely, non-variable subscripts and superscripts that serve as descriptive labels should be set in Roman (upright) type. A thorough review is recommended to ensure this formatting standard is applied consistently throughout the main manuscript and all accompanying Supporting Information.

➤ This has been addressed

The units s⁻¹ and Hz are used inconsistently for rates and frequencies throughout the manuscript. For clarity, the authors should adopt a single, consistent convention and apply it uniformly across the text, figures, tables, and Supporting Information.

➤ This has been addressed

➤

On page 2, line 44, the abbreviation "ES" is used as a plural noun ("ES are..."), which is inconsistent with its singular definition ("a higher energy, excited state (ES)") on line 41. This could create ambiguity for the reader.

➤ This has been corrected to "ESs are"

In Figure 1, the structural representation in sub-panel (f) appears to be drawn at a larger scale compared to those in sub-panels (d), (e), and (g).

➤ Yes indeed, and it was a stylistic choice to show a base-pair and the adjacent bases from both 5' and 3' ends

In the Figure 1 caption, the description "The position of the cross-peaks (in Hz)" is ambiguous. It is not clear what this position is referenced against.

- It has been changed to “The position of the cross-peaks (in Hz, **relative to the diagonal signal**)”

On page 4, line 114, the term "R1ρ" should be correctly formatted with an italicized 'R' and a '1ρ' subscript.

- This has been addressed

The manuscript inconsistently uses both uppercase (X, Y, Z) and lowercase (x, y, z) letters to denote Cartesian axes. For clarity and consistency, a single convention should be adopted and applied uniformly throughout the text and figures.

- The Cartesian axes are now denoted in uppercase across the manuscript

On page 9, line 237, the phrase "these interactions" is ambiguous. For improved clarity, please specify that this refers to the dipolar interactions between the aforementioned proton pairs.

- The sentence is changed to “Given their ~ 3.9 Å distance²⁹, **the dipolar interactions between the proton pairs** minimally influence the on- and off-resonance ^1H $R_{1\rho}$ rates (Figure 2c and 2d).” This section was moved to SI since the NMR for easier access for the general audience.

The caption for Figure 3 should be expanded for clarity by defining "red. χ^2 " (reduced chi-squared). Additionally, the "Residuals" plot is presented without any explicit mention or discussion in the main text.

- This sentence is added to the caption of figure 3: “**Reduced χ^2 (red. χ^2) is denoted on the on-resonance plot and the associated residuals for both on- and off-resonance are shown below the plots.**”

On page 12, line 304, the notation " $^{\text{N}}\text{H}3/1$ " is used without definition. This specialized terminology is potentially ambiguous.

- It is mentioned as Imino $^{\text{N}}\text{H}3/1$, which is the standard notation for the imino protons. For more clarity it has now been changed to “**Imino proton ($^{\text{N}}\text{H}3/1$) chemical shifts showed that in the GS conformation**”

On page 12, line 306, the phrases "lower chemical shift" and "higher chemical shift" are imprecise. For improved technical accuracy and adherence to standard NMR conventions, it is strongly recommended to use the more specific terms "upfield shift" and "downfield shift", respectively. This terminology should be used consistently throughout the manuscript.

- This has been changed to “ **$m^1\text{A}16$ is shifted upfield, while in A_2 $c^7\text{A}16$ it is shifted downfield relative to the wild-type DNA (A_2 wt)**”

The symbol $\Delta\delta$ is used inconsistently. It appears in Figures 3 and 4 without definition, while

the caption for Figure 5 provides two different definitions ("Chemical shift difference" and "Relative chemical shift"). For clarity, a single, explicit definition of $\Delta\delta$ should be provided at its first use and applied uniformly throughout the manuscript.

- For the Figure 3 caption the following sentence is added: “**Relative Chemical shifts ($\Delta\delta$ (ppm)) of HG (green) and ES2 (red) with respect to WCF (blue) are shown for each studied atom**”
- For Figure 5 caption following updates are done for panel b and d respectively:
 - “ **$\Delta\delta$ between the ActD-bound (triangles: Bound A; crosses: Bound B) and free A₂ DNA are overlaid with the experimental $\Delta\delta$ associated with ES2 (red).**”
 - “**Comparison of $\Delta\delta$ values, relative to WCF geometry, for non-exchangeable A16 ¹³C^{H8} and A16 ¹³C^{H2} aromatic protons in ES2 models (Model 1 (orange), and Model 2 (brown)) calculated using DFT with the corresponding experimental value of ES2 (red).**”

Reference 10: This citation is incomplete, as it is missing the final page number.

- This has been addressed

Reference 15: The citation to Angew. Chem. with a volume number of 4 and no page numbers for the year 2016 appears to be incorrect. This entry should be verified and corrected.

- This has been addressed

Reviewer #2

The manuscript by Dasgupta et al. presents a significant advancement in the study of DNA dynamics by addressing a key technical limitation of relaxation dispersion NMR. The authors first systematically quantify the effects of cross-relaxation artifacts, demonstrating that they are negligible for protons more than 3 Å apart, which validates the technique for broader use in nucleic acids. Applying this refined method to a model DNA duplex, they discover a previously hidden, transient intermediate state (ES2) that extends the well-known two-state equilibrium between Watson-Crick-Franklin and Hoogsteen base pairs. The study further characterizes this novel state through a combination of chemical modifications, metadynamics simulations, and DFT calculations to propose a structural model. Finally, the biological relevance of this discovery is highlighted by showing that the ES2 conformation is stabilized by the anticancer drug Actinomycin D, suggesting a new role for transient DNA structures in drug-binding interactions. Our expertise is mainly in Molecular Dynamics (MD) simulations, enhanced sampling, and integration with experimental data, including NMR. Therefore, our review will mostly focus on the modeling part of the manuscript. Our overall assessment is that this manuscript presents a compelling story, composed by combining both experimental NMR data and molecular modeling. However, we have some concerns regarding the latter part that we invite the authors to address in a revised version of the manuscript.

➤ We thank the reviewer for the insightful comments.

Major concerns

1. *Our understanding of the pipeline used by the authors to structurally characterize the ES2 state, starting from its experimental chemical shifts, is as follows:*
 - *Enhanced-sampling metadynamics simulations are performed.*
 - *The resulting trajectory is clustered based on conformational similarity to yield distinct structural groups (3D clusters).*
 - *Chemical shifts (CS) are then calculated for a representative structure from each 3D cluster using Density Functional Theory (DFT).*
 - *These calculated CS are subsequently used for a second clustering analysis, which reveals four major groups (CS clusters).*
 - *One of these CS clusters, designated as Model 1, is identified as the preferred structural model for ES2.*

Based on this (hopefully correct) understanding:

a) *It is unclear whether the 3D clusters are homogeneous with respect to their predicted CS, i.e. whether members of the same cluster exhibit similar (predicted) CS values. While we acknowledge the computational expense of DFT calculations, the authors should demonstrate, for at least one or two representative clusters, that their members have indeed similar predicted CS. This validation is crucial, as the selection of only the cluster center for CS calculation and subsequent CS clustering may not be representative of the entire conformational ensemble.*

➤ *The overall process, as listed by the reviewer, is correct. We have added a new section in the supporting information as *Structural and chemical shift clustering* to explain the entire procedure and the rationale behind each choice. To further assess the homogeneity of the clusters, we performed AFNMR-based chemical shift prediction on an additional one third of randomly selected members of each*

cluster. We added supporting Figure S12a to clarify the homogeneity of the clusters with respect to their chemical shifts depicting a subset of the calculations. The complete result is deposited in the Zenodo repository (<https://doi.org/10.5281/zenodo.17155220>)

b) *Similarly, after clustering in the CS space, the authors should assess whether the resulting CS clusters are structurally homogeneous. A straightforward approach would be to project the centers of all 3D clusters onto the 2D free-energy surfaces in Figure S12, color-coding each point according to its assigned CS cluster. This visualization would make it immediately apparent whether members of the same CS cluster are structurally related or are instead scattered across disparate regions of the conformational space.*

➤ The representation of the structural homogeneity is depicted by the error bars on the chemical shift-selected clusters in Figure S13 (previous S12). We missed to mention this in the original caption of Figure 5 and S12 (now updated to figure S13). We have added the following sentence in the caption.

“Error bars on the chemical shift-based cluster represents 1 SD among the cluster members along the CVs. It is observed that the chemical shift clusters for WCF (black), HG (green) and Model 1 (orange) are structurally homogenous (Figure S12a). For Model 2, the error bars are larger and show some structural inhomogeneity, however, majority of these structures are localised in its unique higher energy local minimum.”

In the main text Figure S12 is referred in the 4th paragraph of Structural models of ES2 as follows:

“The homogeneity of the clusters in the chemical shift space and the convergence of the metadynamics simulation is shown in Figure S12.”

c) *The ES2 excited state is expected to correspond to a local, high-energy minimum on the free-energy surface, likely represented by an ensemble of conformations in rapid exchange that gives rise to a population-averaged experimental CS. In light of the two preceding points, we find the strategy used by the authors for the structural characterization of ES2 not entirely convincing. In our view, a more rational approach would be to identify one or more conformational clusters located in local, high-energy minima from the MD simulations and then determine if these clusters, either individually or as an ensemble average, match the observed ES2 chemical shifts. This contrasts with the current method of clustering in CS space, which risks grouping structurally diverse models from different parts of the conformational landscape. One possible direction would be to exclude the 3D clusters corresponding to the canonical Watson-Crick-Franklin and Hoogsteen states and then apply a Maximum Entropy (MaxEnt) reweighting to the remaining clusters to find the ensemble that best reproduces the experimental ES2 data.*

➤ We thank the reviewer for highlighting the ensemble nature of the ES2 excited state and for suggesting an alternative workflow based on identifying high-energy conformational basins followed by comparison of their ensemble-averaged chemical shifts with experiment. We have shown in Point (a) that our selected clusters are CS homogenous and previously (also addressed in b) indicated that the clusters have distinct structural parameters between the clusters (and are structural homogenous within themselves), we therefore

believe that the method here employed does produce relevant structural distinct clusters, each of which of course has an underlying ensemble. An additional MaxEnt reweighting would primarily redistribute weights within this basin without altering the identity of the underlying conformations. Moreover, excluding the WCF and HG states a priori would remove kinetically and structurally connected regions that define the full transition pathway and contribute to the relative thermodynamic context of ES2 on the free-energy surface.

A full MaxEnt treatment would also require substantially more extensive chemical-shift calculations to ensure statistically meaningful reweighting across all relevant conformational clusters, which is beyond the scope of the present work. We therefore view our current approach (after more clarification), identifying a thermodynamically localized high-energy basin and comparing its population-averaged chemical shifts with experiment, as a consistent and computationally tractable realization of the reviewer's proposed framework.

d) *Finally, it is unclear why the authors opted for DFT calculations over empirical CS predictors. For proteins, it is generally understood that the accuracy of DFT-based predictors is not necessarily superior to that of empirical methods, such as SHIFTX2. The authors should comment on this choice and: i) report the typical error associated with DFT chemical shift calculations for DNA and compare it to the performance of empirical predictors; and ii) explicitly account for this prediction error when comparing calculated and experimental CS for ES2. We believe that looking for a specific conformation (or ensemble) that matches the experimental CS below the typical error in the CS computation is not a robust validation of the model.*

➤ Unfortunately, we have to disagree with the reviewer's comment that DFT-based predictors are not superior to empirical methods for nucleic acids. Even though we would wish that programs as SHIFTX2 and SHIFTS (1, 2) that are optimized for proteins (and it is known that parameters for proteins cannot be used to predict chemical shifts of nucleic acids) would exist in the same quality for NAs. A key motivator is the proven success of the DFT-based AFNMR method in elucidating conformational ensembles DNA (3, 4). Additionally, a thorough comparison of AFNMR, SHIFTS and SHIFTX2 is reported in the original publication (5), which provides confidence in employing this method to the current study. As far as other non-DFT predictors for nucleic acids are concerned, most of the tools are either defunct or outdated (6 – 9):

We have added a sentence in the second paragraph of Structural models for ES2 to highlight the efficacy of AFNMR method

“The efficacy of the AFNMR method compared to other empirical methods to predict the chemical shift of DNA was reported previously^{28,57,60}”

We have added the information about the typical error from DFT in the caption of Figure S11 as

“Typical error from DFT using the parameters used in the current calculation are ^1H (non-labile) = 0.2 to 0.3 ppm, ^1H (labile) \sim 1 ppm, ^{13}C = 2 to 3 ppm and ^{15}N = 4 to 5 ppm^{20,23-25}”

In addition, we also depict the variation among the ensemble structures after clustering. We think that this is more robust than just reporting prediction error from AFNMR method which is reported to be between 0.2 – 0.4 ppm (5).

This prediction was shown to be similar to that obtained from SHIFTS (5) and is lower than what we observed from the ensemble structures. This suggests that our results give conservative estimate of the chemical shift variation. Nevertheless, it matches the experimental observations within the error bars.

We would also like to reiterate that we are not claiming to assign a model conclusively to ES2, rather when we compare the models obtained from the AFNMR method, it matches the experimental observations of modified A₂ DNA and the $R_{1\rho}$ dispersion results.

In terms of matching the experimental observation and the predicted chemical shift, we opted to rely on the credibility of published work which included rigorously testing (3, 4). Therefore, we think that this comparison provides a starting model that can be tested in follow-up work and either re-affirm or improve the model for ES2.

1. Han, B., Liu, Y., Ginzinger, S. W. & Wishart, D. S. SHIFTX2: significantly improved protein chemical shift prediction. *J Biomol NMR* **50**, 43–57 (2011).
 2. Xu, X.-P. & Case, D. A. Automated prediction of ¹⁵N, ¹³C α , ¹³C β and ¹³C' chemical shifts in proteins using a density functional database. *J Biomol NMR* **21**, 321–333 (2001).
 3. Shi, H. *et al.* Atomic structures of excited state A–T Hoogsteen base pairs in duplex DNA by combining NMR relaxation dispersion, mutagenesis, and chemical shift calculations. *J Biomol NMR* **70**, 229–244 (2018).
 4. Zhou, H. *et al.* Characterizing Watson–Crick versus Hoogsteen Base Pairing in a DNA–Protein Complex Using Nuclear Magnetic Resonance and Site-Specifically ¹³C- and ¹⁵N-Labeled DNA. *Biochemistry* **58**, 1963–1974 (2019).
 5. Swails, J., Zhu, T., He, X. & Case, D. A. AFNMR: automated fragmentation quantum mechanical calculation of NMR chemical shifts for biomolecules. *J Biomol NMR* **63**, 125–139 (2015).
 6. NUCHEMICS, Wijmenga, S. S., Kruithof, M. & Hilbers, C. W. Analysis of ¹H chemical shifts in DNA: Assessment of the reliability of ¹H chemical shift calculations for use in structure refinement. *J Biomol NMR* **10**, 337–350 (1997).
 7. RAMSEY, Frank, A. T., Bae, S.-H. & Stelzer, A. C. Prediction of RNA ¹H and ¹³C Chemical Shifts: A Structure Based Approach. *J. Phys. Chem. B* **117**, 13497–13506 (2013).
 8. Lam, S. L. DSHIFT: a web server for predicting DNA chemical shifts. *Nucleic Acids Res* **35**, W713–W717 (2007).
2. *Our second major concern relates to the methodology and validation of the enhanced-sampling simulations.*
- a. *First, the manuscript lacks a rigorous assessment of simulation convergence, which is essential, particularly given the relatively short 200 ns simulation length. A standard validation would involve presenting time-series plots for each metadynamics CV alongside a block analysis to estimate the statistical error on the 1D or 2D free-energy surfaces. Leveraging the authors' existing 3D cluster analysis, an even more direct approach would be to report the reweighted, unbiased population of each cluster with its associated statistical error.*
- We thank the reviewer for pointing out our failure to clearly state convergence, although the simulations did converge. We have now

included a plot depicting the reweighted, unbiased population of each cluster with its associated statistical error in Figure S12b and S12c. The populations are observed to be stable with small statistical error, indicating convergence of the enhanced-sampling simulations. We have added a section *Structural and chemical shift clustering* in the supporting information detailing the process.

- b. *Second, the rationale for selecting the specific collective variables (CVs) is not completely clear. In particular, two of the chosen CVs (the 'Base Flipping' angle and the 'T9 O4–A16 N1 distance') do not effectively discriminate between the WCF and HG states, questioning their relevance as a CV in metadynamics simulations. Furthermore, Table S11 also reports as CV the 'Centre of Mass C15 – A17' (also with no difference between the WCF and HG states), without clarifying whether this variable was ultimately used in the simulations.*

- We agree that the base-flipping angle and the T9 O4–A16 N1 distance do not, distinguish WCF from HG. However, our aim was not to distinguish WCF and HG but rather: (a) explore extra-helical and partially flipped intermediates along the transition pathway, and (b) improve sampling of conformations that are orthogonal to the canonical WCF – HG reaction coordinate, which the selected CVs are well equipped to do so. We show that the CVs used for metadynamics identify novel wells/paths to additional states. Additionally, the choice was motivated from various studies that have explored the WCF – HG dynamics using metadynamics or enhanced molecular dynamics simulations. We have now made it clear in the Table S11 heading as follows:

“Average values and the standard deviation of the collective variables used for the metadynamics simulations³⁹⁻⁴⁴ for both WCF and HG conformation from unbiased 100ns MD simulation. The WCF – HG transition was probed for A16 in A₂ DNA”

- We thank the reviewer for pointing out the ambiguity. The CV “Centre of Mass C15 – A17” is a remnant of initial exploration. It was not used for the final production run. We have removed it from Table S11.

Minor concerns

1. *In Figure S11, it is unclear what the dots and error bars of the CS calculated with DFT represent.*

- We have added the following description at the end of the figure caption S11:
“In the DFT-predicted $\Delta\delta$ (ppm), dots represent the mean difference of mean chemical shift between WCF and HG conformations and error bars the standard deviation via error propagation, respectively, obtained from the all the structures defining the ensemble of each conformation.”

2. *Coloring in Figure S11d is a bit confusing, as red and brown look very similar to us.*

- We have updated the circles for the DFT calculated relative chemical shift for Model 1 and Model 2 with respect to the WCF conformation as filled circle to increase contrast.
3. *Figure S11d should report the typically error in DFT calculations of CS to appreciate whether the observed differences are significant given this typical error.*
- We have updated the figure caption mentioning the typical DFT error on the calculated chemical shift as:

“Typical DFT error using the parameters used in the current calculation are ^1H (non-labile) = 0.2 to 0.3 ppm, ^1H (labile) \sim 1 ppm, ^{13}C = 2 to 3 ppm and ^{15}N = 4 to 5 ppm^{20,38-40}”
4. *In the caption of Figure S12, it is unclear what an “Energy-weighted” free energy is.*
- We have clarified this by substituting the “Energy-weighted” with “Two-dimensional free energy surface...”

REVIEWERS' COMMENTS

Reviewer #1 (Remarks to the Author):

The authors have adequately addressed all points raised in the initial review, with one minor exception regarding the reference list.

While Reference 15 was corrected, the identical paper is duplicated as Reference 29.

- We thank the reviewer for the comment. The issue with the references is now corrected and reference 29 was removed.

Reviewer #2 (Remarks to the Author):

The authors addressed satisfactorily all my previous concerns and revised the manuscript accordingly.

I would like just to point out that in my original report I never stated that "DFT-based predictors are not superior to empirical methods for nucleic acids", as remarked by the authors in their rebuttal. Actually, I was pointing out that *for proteins*, it is generally understood that the accuracy of DFT-based predictors is not necessarily superior to that of empirical methods. With my statement, I was inviting the authors to make a comment regarding DFT vs empirical predictors in nucleic acids, as I was not familiar (and maybe the reader as well) with their performances for these molecules. In any case, the authors ultimately addressed this point by citing previous works where these forward models were extensively tested on nucleic acids.

Reviewer #3 (Remarks to the Author):

- We thank the reviewer for the clarification and are happy to have provided all the required information. We appreciate the possibility to clarify to the general reader the state of the art of DFT-based predictors vs empirical methods for NAs.